# CRISPR-free RNA base editing mediated PTC-readthrough restores hearing in mice with *Otof* nonsense mutation

Hanxiao Sun[1,11], Qi Teng[2,11], Wenqing Liu[3,4,11], Rui Guo[2,11], Menghua Li[2,11], Wei Xiong[2], Qiang Huang[1], Qianru Yu[2], Nan Luo[1], Yang Li[2,5], Jinghui Song[1], Shusheng Gong[2,6], Xi Shi[7] ✉, Chengqi Yi[1,8,9,10] ✉ & Ke Liu[2,5,6] ✉

The gene therapy achieved by AAV-mediated otoferlin-overexpression is an effective therapeutic strategy for congenital deafness. However, achieving its physiological and endogenous patterns of expression remains challenging. Here, we generate the homologous mutation *Otof c.1315 C > T* (p.R439*), equivalent to *OTOF c.1273 C > T* (p.R425*) found in humans with profound deafness, to create a nonsense mutation-induced deaf mouse model. We then deliver the 'RESTART v3' system, which is a CRISPR-free RNA base editor for nonsense mutation suppression, into the cochlea of the mice. We achieve physiological otoferlin expression, and the edited premature termination codon is reverse-mutated to the original amino acid. We observe significant hearing restoration and enhancement of the behavioral auditory startle reflex. Thus, our study presents a successful RNA editing strategy to significantly restore hereditary deafness in mice carrying the specific *Otof* nonsense mutation, which holds great promise for future clinical translation.

Congenital hearing loss occurs in approximately1–3 out of 1000 newborns worldwide[1–3]. Approximately over 30,000 deaf infants are born in China annually[4]. Congenital hearing loss significantly impairs the communication and cognitive development of children. Over 50% of non-syndromic congenital profound deafness cases are associated with genetic deficits[5,6], most commonly autosomal recessive (DFNB) forms[7]. The currently available cochlear implants are insufficient to meet the needs of deaf children in both sound recognition and perception, despite being considered the best technical achievement thus far for treating hearing loss[8–11]; thus, there is an urgent demand for genetic manipulation in situ and live gene editing to treat deaf children.

Hundreds of genes that contribute to hereditary hearing impairment have been identified, among them *OTOF* is one of the most extensively investigated genes[12,13]. *OTOF*-related deafness is characterized by the phenotypes of prelingual non-syndromic auditory neuropathy, resulting in severe-to-profound bilateral deafness and normal morphology of cochlear hair cells[13]. In early 2024, the first study of successful use of gene therapy for deafness in patients with *OTOF* mutations was reported[14]. However, AAV or dual-AAV, a common biological tool for inner ear delivery of gene therapy, has insufficient carrying capacity, particularly for delivering large genes[15]. Additionally, dual-AAV vector approaches are limited by their low reconstitution efficiency, production of alien proteins, and flexibility in

[1]The National Key Laboratory of Gene Function Studies and Manipulation, School of Life Sciences, Peking University, Beijing, China. [2]Department of Otolaryngology Head and Neck Surgery, Beijing Friendship Hospital, Capital Medical University, Beijing, China. [3]School of Life Sciences, Tsinghua University, Beijing, China. [4]Tsinghua-Peking Joint Center for Life Sciences, Tsinghua University, Beijing, China. [5]Beijing Clinical Research Institute, Beijing, China. [6]Clinical Center for Hearing Loss, Capital Medical University, Beijing, China. [7]Academician Workstation of Hainan University (Sanya), School of Pharmaceutical Sciences, Hainan University, Haikou, Hainan, China. [8]Peking-Tsinghua Center for Life Sciences, Peking University, Beijing, China. [9]Department of Chemical Biology and Synthetic and Functional Biomolecules Center, College of Chemistry and Molecular Engineering, Peking University, Beijing, China. [10]Beijing Advanced Center of RNA Biology (BEACON), Peking University, Beijing, China. [11]These authors contributed equally: Hanxiao Sun, Qi Teng, Wenqing Liu, Rui Guo, Menghua Li. ✉e-mail: shixi2023@hainanu.edu.cn; chengqi.yi@pku.edu.cn; liuke@ccmu.edu.cn

split site selection[16]. The *Otof* splitting site is critical for recombination efficiency; however, as different splitting sites were used in different studies, there is a lack of consensus on current gene therapies[17,18]. Additionally, gene therapy carries potential risks, including uncontrollable, over-, or ectopic expression[19,20]. In fact, otoferlin-overexpressing AAV vectors have been found in the central nervous system (CNS) and liver, in addition to their predominant distribution in the inner ear[17,21]. The distribution of AAV in other organs might be caused by the cochlear aqueduct, which is a communication channel between the cerebrospinal fluid (CSF) and perilymph[17,21]. Hence, these limitations of the current gene therapies result in an urgent need for alternative strategies. Thus, RNA editing, an emerging approach, could shed light on technical solutions to overcome the limitations of gene therapies[22–24].

RNA editing tools target endogenous RNA transcripts without the need to overexpress a large target gene, as is often encountered in AAV-mediated DNA editing[22–25]. Several adenine to inosine (A-to-I) and cytosine to uracil (C-to-U) RNA base editors have been developed[22,23,26]. The A-to-I RNA editor, based on the deamination activity of ADAR proteins, is currently the predominantly used RNA-editing tool. Previously, several researchers, including our group, have independently developed a programmable RNA base editor RESTART to suppress premature termination codons (PTCs)[27,28]. RESTART introduces a pseudouridine (Ψ) modification at the PTC site and enables protein read-through, thereby allowing the generation of a full-length protein[28]. Recently, we upgraded this tool, named RESTART v3[29], which uses near-cognate tRNA to greatly enhance the PTC read-through efficiency. In our previous study using primary cells, the RESTART v3 system efficiently enabled protein expression, leading to the rescue of cellular phenotypes. Furthermore, it demonstrated minimal off-target effects and was sufficiently small for viral delivery[29,30]. However, the advantages of the RESTART system have not been demonstrated in vivo, particularly in an animal model of congenital hearing loss.

In this study, we generate a profound hearing loss mouse model by introducing the homologous mutation *Otof c.1315 C > T* (p.R439*), equivalent to *OTOF c.1273 C > T* (p.R425*) in human, which has been previously identified in three human families[31–33]. We deliver RESTART v3 into the inner ear of *Otof c.1315 C > T* (p.R439*) mice at postnatal day 28 (P28) to test hearing rescue. The results

reveal significant restoration of hearing, significant enhancement of behavioral auditory startle reflex (ASR), and normalization of otoferlin expression in the cochlear inner hair cells in the treated mice. Therefore, our study demonstrates that the RESTART v3 system achieves an efficient and precise read-through at the PTC site (*Otof c.1315 C > T*), incorporating the original amino acid (arginine) at the nonsense mutation position and generating the full-length, functional otoferlin protein, thereby alleviating site-specific humanization congenital deafness. In addition, it also be a common strategy to other genetic disease[29], and could cover nearly 20% PTC sites found in patients with *OTOF* deficiency[34]. These findings hold importance not only for the academic community but also have the potential to contribute substantially to advancements in clinical practices and therapeutic approaches.

## Results

### Generation of the *Otof c.1315 C > T* (p.R439*) mouse model

To validate the in vivo RNA-editing efficiency, we generated a homology *Otof c.1315 C > T* (p.R439*), equivalent to *OTOF c.1273 C > T* (p.R425*) in human, deaf mouse model because the mutant site has been identified to cause deafness in human families (Supplementary Fig. 1). We used the CRISPR-Cas9 system guided by specifically designed sgRNAs flanking exon 14 and a donor oligo DNA carrying a T mutation to generate the mouse mutation (Fig. 1a, b). Importantly, a homology alignment of the *Otof* CDS sequence showed a 100% homology region (32 bp) around the mutated site between Mus and Homo, representing a native site-specific humanization *Otof c.1315 C > T* (p.R439*) mouse model (Fig. 1c).

### Functional and immunostained evaluation of the native *Otof c.1315 C > T* (p.R439*) mice

To assess the auditory function of *Otof c.1315 C > T* (p.R439*) mice, we examined the auditory brainstem response (ABR) of broadband clicks and tone bursts at 8 weeks old (Fig. 2a, b), the results showed profound hearing loss in these *Otof c.1315 C > T* (p.R439*) mice. Next, we evaluated the expression of otoferlin in IHCs, dissected mice cochlea and harvested the basilar membrane for immunofluorescent staining (Fig. 2c). The images showed nearly absence of otoferlin expression in the *Otof c.1315 C > T* (p.R439*) mice compared with the same-aged wild-type (WT) mice (Fig. 2d). In addition,

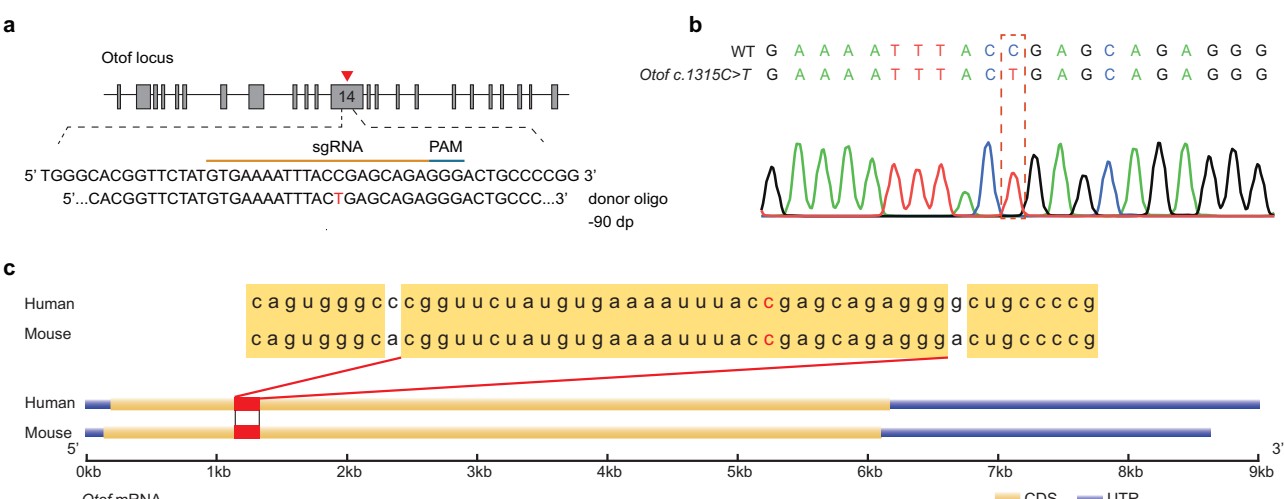

**Fig. 1 | The generation of the native *Otof c.1315 C > T* (p.R439*) mouse model. a** sgRNA design to construct the *Otof c.1315 C > T* (p.R439*) mouse model. *Otof*-targeted sgRNA contains a complementary C pairing with the 1315 C > T located in exon 14 of the *Otof* gene. **b** Sanger sequencing indicated the success of the construction of *Otof c.1315 C > T* (p.R439*) mouse model. **c** Homology alignment analysis between Mus (Mouse) and Homo (Human) *Otof* transcripts V1, and the frame showing the 100% homology region that includes the site of *Otof c.1315 C > T*. The marked c base corresponds to position c.1315 in the mouse *Otof* gene and c.1273 in the human *OTOF* gene. CDS: Coding Sequence. UTR: Untranslated Region.

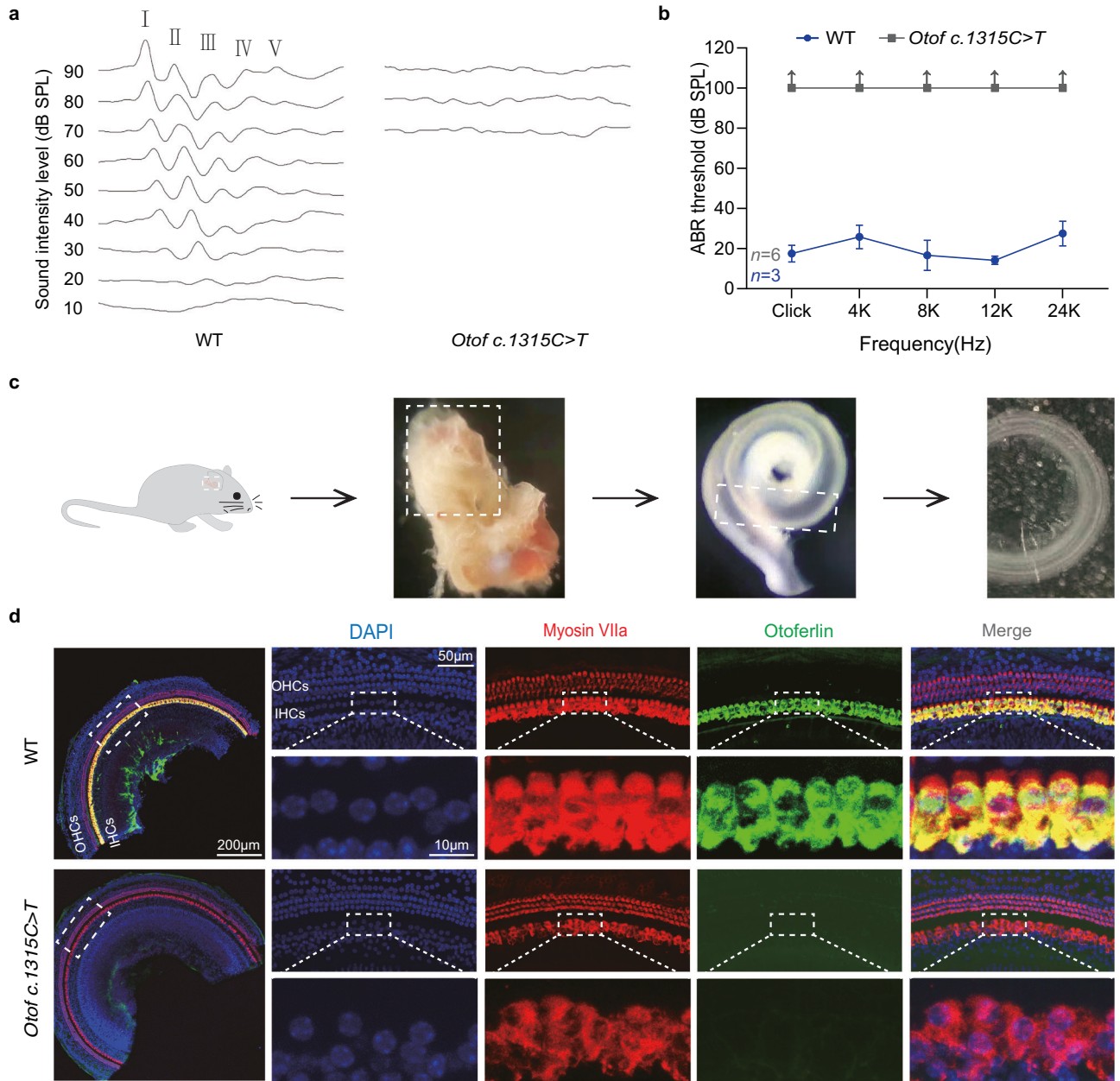

**Fig. 2 | Functional and immunostained evaluation of *Otof c.1315 C > T* (p.R439*) mice. a** No ABR waveforms were elicited in the *Otof c.1315 C > T* (p.R439*) mice compared with the WT mice. **b** Statistical analysis of ABR thresholds between the *Otof c.1315 C > T* (p.R439*) (*n* = 3 biological samples) and WT mice (*n* = 6 biological samples), significant elevation of ABR thresholds was seen in the *Otof c.1315 C > T* (p.R439*) mice compared with the WT mice. Data are represented as the mean ± SD. **c** A schematic diagram for the harvest of cochlear basilar membrane.

**d** The whole mount staining showed absence of otoferlin expression in the *Otof c.1315 C > T* (p.R439*) mice, whereas there was normal otoferlin expression in the WT mice. The white frames indicate the enlarged region of IHCs. (Myosin VIIa, red; otoferlin, green; DAPI, blue.). Scale Bars = 200 μm, 50 μm, 10 μm. This experiment was conducted with three independent replicates. IHCs: inner hair cells; OHCs: outer hair cells. Source data are provided as a Source Data file.

we calculated the quantity of cochlear hair cells, and found no significant changes between the *Otof c.1315 C > T* (p.R439*) mice and the WT mice (Supplementary Fig. 2). The results demonstrated a lack of otoferlin expression and profound hearing loss in the native *Otof c.1315 C > T* (p.R439*) mice (Fig. 2).

## CRISPR-free RNA editing RESTART-mediated *Otof* PTC-readthrough

Our recent work shows that RESTART v3, which overexpresses the natural tRNA-R-TCT-1-1 to repair the UGA stop codon, not only enhances the readthrough efficiency but also specifically decodes it as arginine[29]. We envisioned that such a precise editing approach should

generate the exact wild-type otoferlin protein from the *Otof* gene (*c.1315 C > T*, p.R439*) and preserve the integrity of the OTOF protein's structure and function (Fig. 3a).

In this study, we selected gACA19 and gACA36 backbones due to their robust secondary structural stability (Fig. 3b and Supplementary Fig. 3a). We evaluated RESTART mediated readthrough efficiency through a fluorescent reporter gene system, juxtaposing a 21-nucleotide sequence near the *Otof* PTC site between mCherry and EGFP protein (Fig. 3c, d and Supplementary Fig. 3b). With our previous chemical-assisted PRAISE method for precise pseudouridine (Ψ) quantification, we tested RNA editing efficiency across different RESTART versions (Fig. 3e)[35]. Employing gACA19 alone (RESTART v1)

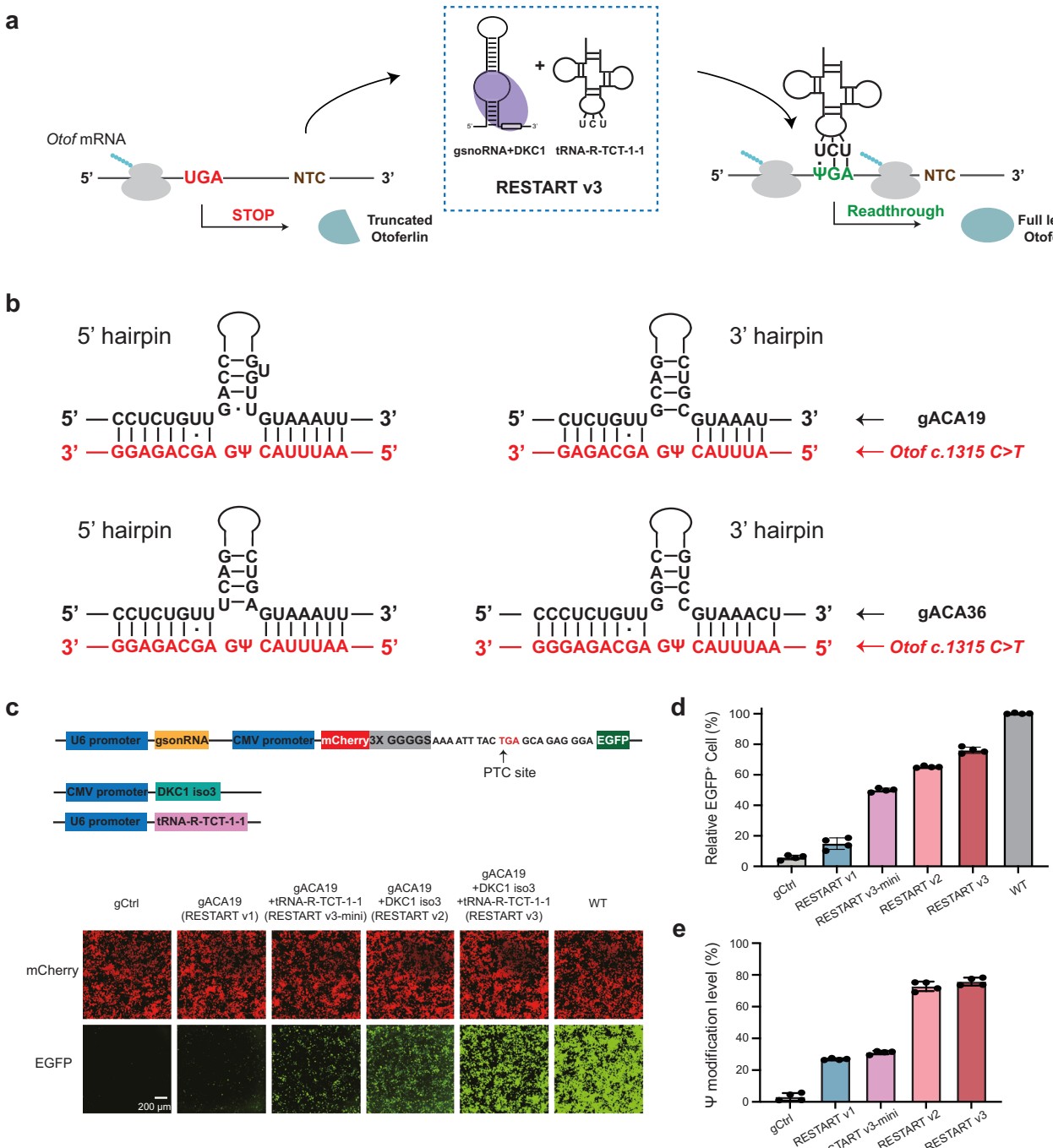

**Fig. 3 | CRISPR-free RESTART v3 achieved efficient readthrough of the *Otof* premature termination codon (PTC) site. a** Schematics of the RESTART v3 design. Schematics showing that gsnoRNA and DKC1 isoform3 (iso3) enable pseudouridylation on mouse *Otof* (*c.1315 C > T*, p.R439*) PTC site and tRNA-R-TCT-1-1 can recognize ΨGA site, achieving efficient readthrough of PTC site. NTC stands for normal termination codon. **b** Complementary regions between gsnoRNAs (black) and target sites in *Otof* transcripts (red). **c** Top, schematics of gsnoRNA-PTC-reporter, DKC1 iso3 and tRNA-R-TCT-1-1 constructs. Bottom, the indicated gACA19-

PTC-reporter, DKC1 iso3, and tRNA-R-TCT-1-1 were transfected into HEK293T cells. Representative fluorescence images of cells. Scale Bars, 200 μm. This experiment was conducted with four independent replicates. **d**, **e** The indicated gACA19-PTC-reporter, DKC1 iso3, and tRNA-R-TCT-1-1 were transfected into HEK293T cells. Bar plot showing the relative fraction of EGFP-positive cells (**d**). Bar plot showing different modification level across different RESTART versions (**e**). *n* = 4 biological replicates. Data are represented as the mean ± SD. Source data are provided as a Source Data file.

resulted in a 27% modification level at the PTC site, showing a modest EGFP signal. Incorporating tRNA-R-TCT-1-1 (RESTART v3-mini) significantly improved EGFP expression without altering Ψ modification levels. The co-overexpression of gACA19 and DKC1-iso3 (RESTART v2) propelled the modification rate to 75% with a corresponding increase in the EGFP expression level compared with RESTART v1 (the relative fraction of EGFP-positive cells and intensity increased to 66% and 9.8%,

respectively). Finally, the RESTART v3 system, which combines tRNA-R-TCT-1-1 with the RESTART v2 system, led to a more substantial increase in EGFP expression (the relative fraction of EGFP positive cells and intensity increased to 75% and 29%, respectively, Fig. 3d and Supplementary Fig. 3b). gACA36's performance was comparable to gACA19 in terms of editing efficiency and protein expression (Supplementary Fig. 4).

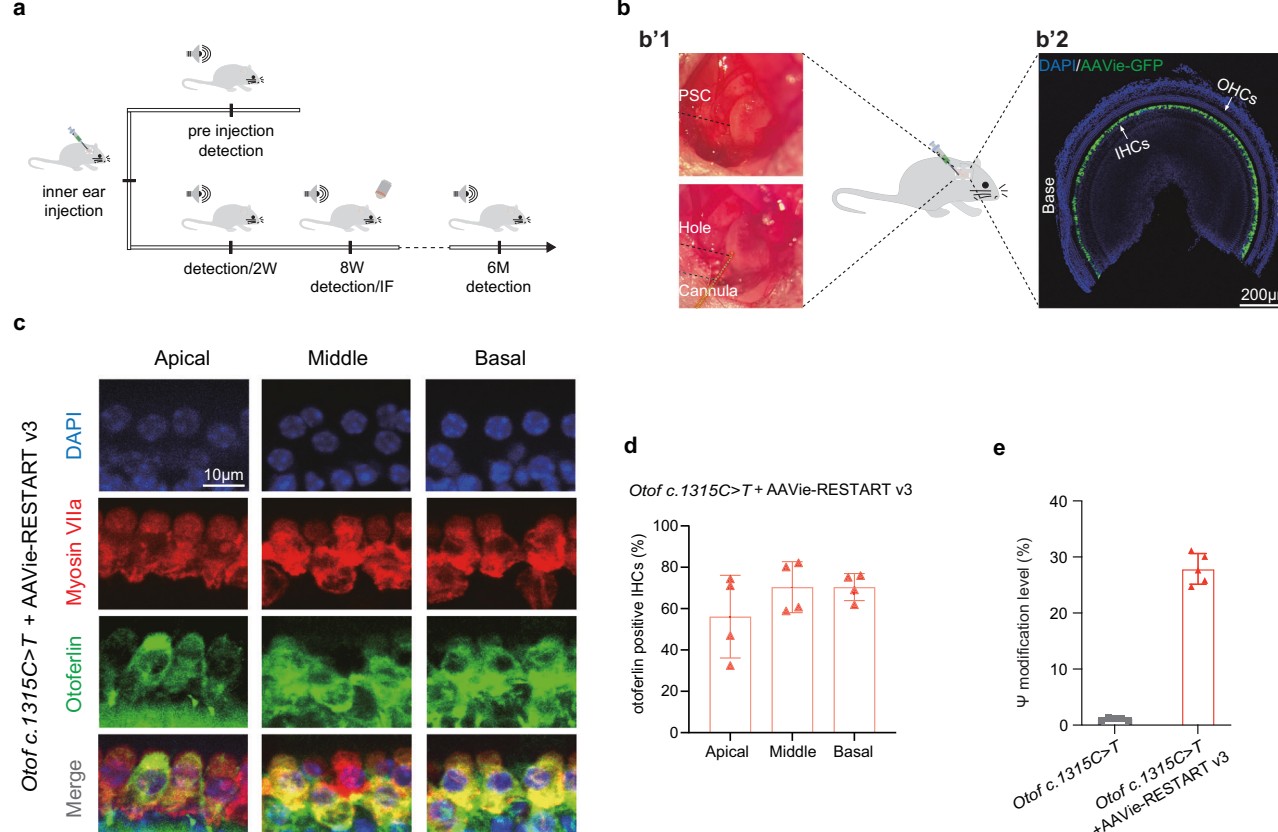

**Fig. 4 | AAV-RESTART v3 rescues otoferlin expression in the *Otof c.1315 C > T* (p.R439\*) mice. a** Schematic diagram for verification of RNA editing in the *Otof c.1315 C > T* (p.R439\*) mice. Before the injection of the virus, the hearing detection was performed. The AAV was delivered into the inner ear of mice via the posterior semicircular canals (PSC) route; hearing was examined every 2 weeks since the virus delivery for 8 consecutive weeks. The last hearing test was performed at the 6th month after the injection, and immunostaining was performed at the 8th week after the injection. **b–b'1** The images of the surgical operation of PSC. **b–b'2** Representative confocal microscopy image of transfection efficiency of AAVie serotype virus in inner ear, indicating a highly specific and transfected efficiency in IHCs, no immunostained signals were seen in OHCs. IHCs: inner hair cells; OHCs: outer hair cells; Scale Bars, 200 μm. **c** Confocal images of the *Otof c.1315 C > T* (p.R439\*) mice cochlea after the injection, robust otoferlin expression (green) can be found in the IHCs from apical to basal turns, the cochlear hair cells were labeled with myosin VIIa (red), cell nuclei were stained with DAPI (blue). Scale Bars, 10 μm. **d** Quantification of the proportion of otoferlin positive IHCs in the treated *Otof c.1315 C > T* (p.R439\*) mice (*n* = 4 biological replicates). Data are represented as the mean ± SD. **e** Nearly 30% pseudouridine modification efficiency achieved at the targeted premature termination codon (PTC) site in the treated *Otof c.1315 C > T* (p.R439\*) mice (*n* = 5 biological replicates). Data are represented as the mean ± SD. Source data are provided as a Source Data file.

To further test the efficacy of RESTART, we engineered additional constructs by containing the full-length *Otof* gene carrying *c.1315 C > T* (p.R439\*) nonsense mutation (Supplementary Fig. 5a). Upon introduction of the RESTART tool, we observed efficient pseudouridine modification level at the PTC site (~66%, Supplementary Fig. 5b). Collectively, our experiments validated the feasibility of RESTART v3 as a viable tool for PTC-readthough for the *OTOF* gene.

### AAV-mediated delivery of RNA editing RESTART v3 rescues otoferlin expression in the *Otof c.1315 C > T* (p.R439\*) mice

For in vivo delivery of RESTART v3, we applied AAVie serotype which has high transfection efficiency and specific expression in IHCs[36] (Fig. 4a, b). AAVie-RESTART v3 was delivered into inner ear of *Otof c.1315 C > T* (p.R439\*) mice via the posterior semicircular canal (PSC) at P28 (Fig. 4a, b). Hearing detection was conducted in every two weeks for 8 consecutive weeks, the last hearing test has been performed at the 6th month after the injection. After the hearing detection, the cochlear samples of both the treated *Otof c.1315 C > T* (p.R439\*) mice and the untreated mice were immunostained to estimate the otoferlin expression (Fig. 4a). At the 8th week after injection, the results showed a robust otoferlin expression in the most of IHCs from apical to basal turns, the immunofluorescence signals for otoferlin and myosin VIIa

were highly overlapping on the IHCs (Fig. 4c). We next calculated the proportion of otoferlin signals positive IHCs, and more than 70% of the IHCs were found to express otoferlin in the middle and basal turns, a slightly reduced proportion of positive otoferlin signals appeared at the apical turn (Fig. 4d). Furthermore, our study showed an approximate 30% pseudouridine modification efficiency at the targeted PTC site (Fig. 4e). Consistent with our previous study demonstrating that RESTART-mediated PTC readthrough acts synergistically with nonsense-mediated mRNA decay (NMD) inhibition[29], we also observed RESTART v3 treatment could stabilize the *Otof c.1315 C > T* transcripts, which are sensitive to NMD (Supplementary Fig. 6a). Thus, these results indicated that RESTART v3 mediates efficient PTC editing and restores endogenous otoferlin expression in IHCs.

### RNA editing RESTART v3 can significantly restore the hearing of the *Otof c.1315 C > T* (p.R439\*) mice

To explore whether the RESTART v3 can restore auditory function in *Otof c.1315 C > T* (p.R439\*) mice, we then examined the ABR thresholds across the frequencies for 6 month after injection (Fig. 5). The RESTART v3 treated mice revealed a visible ABR waveforms with normal characteristics at all detected frequencies, in which wave I−V were identifiable, whereas, there were no identifiable ABR waveform elicited

in the untreated mice (Fig. 5a). Additionally, we found a prominent summating potential (SP) before wave I, which was normally evoked by the depolarized OHCs[13,37]. In this study, the treated *Otof c.1315 C > T* (p.R439*) mice showed average about 60 dB ABR thresholds at the tested frequencies (Fig. 5b). Moreover, we estimated the latency and amplitude of ABR wave I using the 90/80 dB sound intensity in the treated *Otof c.1315 C > T* (p.R439*) mice; also, we found normalized properties including a prolonged latency (Fig. 5c′1) and reduced amplitude (Fig. 5c′2) in response to the decreased sound stimuli in the *Otof c.1315 C > T* (p.R439*) mice (Fig. 5c).

### Auditory startle reflex (ASR) analysis of RNA editing RESTART v3 treated *Otof c.1315 C > T* (p.R439*) mice

To further confirm whether the treated *Otof c.1315 C > T* (p.R439*) mice achieve a convincing restoration of hearing, we utilized ASR, a behavior analysis, to examine the recovery of hearing function[38]. The tested mice could present muscles tension and jumping once hearing a sudden loud sound (Supplementary Movie 1 and 2), and the gravity sensor under the detection platform can calculate these changes by gravitational acceleration (Fig. 5d). In this study, we found stable ASR waveforms elicited in the treated *Otof c.1315 C > T* (p.R439*) mice, the amplitude of startle reflex in the treated mice was slightly lower than that of the WT mice. However, no visible ASR waveform can be seen in the AAVie-GFP injected *Otof c.1315 C > T* (p.R439*) mice (Fig. 5e). Further, the stable ASR occurred at high frequency in the treated *Otof c.1315 C > T* (p.R439*) mice, by contrast, there was no ASR appeared at the untreated mice (Fig. 5f), also, there is statistical difference in ASR waveform amplitude in the treated *Otof c.1315 C > T* (p.R439*) mice compared with the untreated mice (Fig. 5g). In addition, we also performed ASR test at the 6th month after the injection, and the treated mice showed a similar behavior performance compared to the mice at the 8th week since the injection (Fig. 5f, g).

### Specificity of RESTART v3

Off-target evaluation is critical for RNA editing technologies. We evaluated the specificity of RESTART v3 comprehensively under in vitro conditions. First, we performed whole-transcriptome Ψ sequencing to identify off-target editing by RESTART v3. Only around thirty off-target sites were identified in the mRNA of cell line treated with RESTART v3; these sites were enriched in the coding sequence (CDS) regions and avoided normal termination codons (Supplementary Fig. 6b). Moreover, the off-target sites displayed sequence motifs complementary to the gsnoRNA sequence (Supplementary Fig. 6c), indicating that the off-target effects are dependent on gsnoRNA—this finding consistent with previous studies[28,29]. Next, the quantitative proteome analyses demonstrated that RESTART v3 enabled efficient incorporation of arginine (~99.98%) at *Otof c.1315 C > T* PTC site, and no detectable perturbation of global protein abundance, including genes enriched for AGA codons (Fig. 6a, b and Supplementary Fig. 6d).

To explore the potential off-target editing and the subsequent effects of RESTART v3 in vivo, we conducted transcriptome-wide sequencing to quantify Ψ sites across the mouse cochlear tissues, collected four weeks after AAV injection. Analysis revealed over 900 endogenous Ψ modification sites in the cochleae of untreated mice, which are enriched in the CDS and 3′UTR regions (Supplementary Fig. 6e, and Supplementary Data 1). Such a distribution pattern is reminiscent of the Ψ profile we previously obtained in several cell lines[35,39]. Post AAVie-RESTART v3 injection, these sites maintained consistent stoichiometry (Fig. 6c). Comparative analysis between RESTART v3-treated and untreated samples revealed no new Ψ modifications induced by RESTART v3 treatment, suggesting no detectable off-target edits under the current in vivo condition. Such specificity appears to outperform our previous observations under in vitro condition, where we found tens of off-target Ψ sites (Supplementary

Fig. 6b). This could be due to the different transfection efficiencies between cultured cell lines and the cochlea. To further look into specificity of RESTART v3, we bioinformatically predicted the top 50 regions resembling gsnoRNA complementary sequences, given that off-targets induced by RESTART v3 are dependent on gsnoRNA sequence. We did not find any aberrant modification at these loci (Supplementary Fig. 6f, and Supplementary Data 2), again suggesting no detectable off-target editing sites. The absence of off-target effects in vivo and the presence of only around thirty off-target Ψ sites in vitro for RESTART v3 demonstrated its high specificity.

To comprehensively evaluate the molecular safety profile of RESTART v3, we conducted ribosome profiling (Ribo-seq) on cochleae from untreated *Otof c.1315 C > T* mice and mice treated with RESTART v3 for three months. We first quantified the ribosome-protected fragment (RPF) density within 3′ UTR regions across all transcripts. We observed no enrichment of downstream 3′ UTR RPF signal in RESTART v3–treated samples relative to untreated controls (Supplementary Fig. 7a), indicating no detectable increase in off-target readthrough at normal stop codons (NTCs). At the per-transcript level, we calculated the ribosome readthrough score (RRTS). We detected 15 and 17 potential readthrough events in untreated and treated samples, respectively. 6 events are shared between groups, while 9 and 11 events appear to be present in untreated and treated samples, respectively (Supplementary Fig. 7b). For the 11 events, we did not detect Ψ at the corresponding stop codons, and these sites lacked complementarity to the guide snoRNA. These features argue against RESTART-mediated editing and are consistent with endogenous and low-frequency readthrough. Taken together, these analyses reveal no measurable increase in off-target readthrough at NTCs following RESTART treatment. We further analyzed globe translational efficiency and found no significant alterations after treatment (Supplementary Fig. 7c). Collectively, the Ribo-seq safety readouts support the high specificity of RESTART v3 and minimize concerns regarding proteome-wide perturbations.

### Global transcriptome and tRNA landscape remain stable after RESTART v3

To further evaluate the potential impact of RESTART v3 on global transcription, we performed RNA-seq experiments on cochlear tissues from mice both before and after treatment. Our data showed that RESTART v3 does not influence overall RNA expression levels (Fig. 6d), similar to our previous observations in cell lines[29].

To quantify episomal tRNA-R-TCT-1-1 expression, we conducted tRNA sequencing (tRNA-seq) on cochlear samples collected at two weeks (onset of AAV expression) and six weeks (stable AAV expression) post-injection of AAVie-RESTART v3. R-TCT-1-1 levels were elevated at 2 weeks and further increased by 6 weeks, consistent with the maturation and stabilization of AAV-driven expression. By 6 weeks, R-TCT-1-1 abundance reached approximately 1.5-fold over untreated cochleae (Supplementary Fig. 7d). This magnitude aligns with our prior cell-based studies in HEK293T cells, where exogenously expressed R-TCT-1-1 typically reached ~2-fold over wild-type cells under highly efficient delivery[29].

We next assessed the impact of tRNA-R-TCT-1-1 overexpression on endogenous tRNAs. Grouping tRNAs by anticodon identity revealed no significant changes across tRNA classes after treatment (Fig. 6e). Further stratification by isodecoder families likewise showed no global perturbations (Supplementary Fig. 7e). Within the arginine isoacceptor family, only the introduced tRNA-R-TCT exhibited elevated abundance; no significant changes were observed in other Arg-isoacceptor tRNA (Supplementary Fig. 7f). Together, these results indicate that RESTART v3 drives specific expression of the engineered near-cognate tRNA without measurable alterations to the endogenous tRNA pool, supporting its high specificity and favorable safety profile.

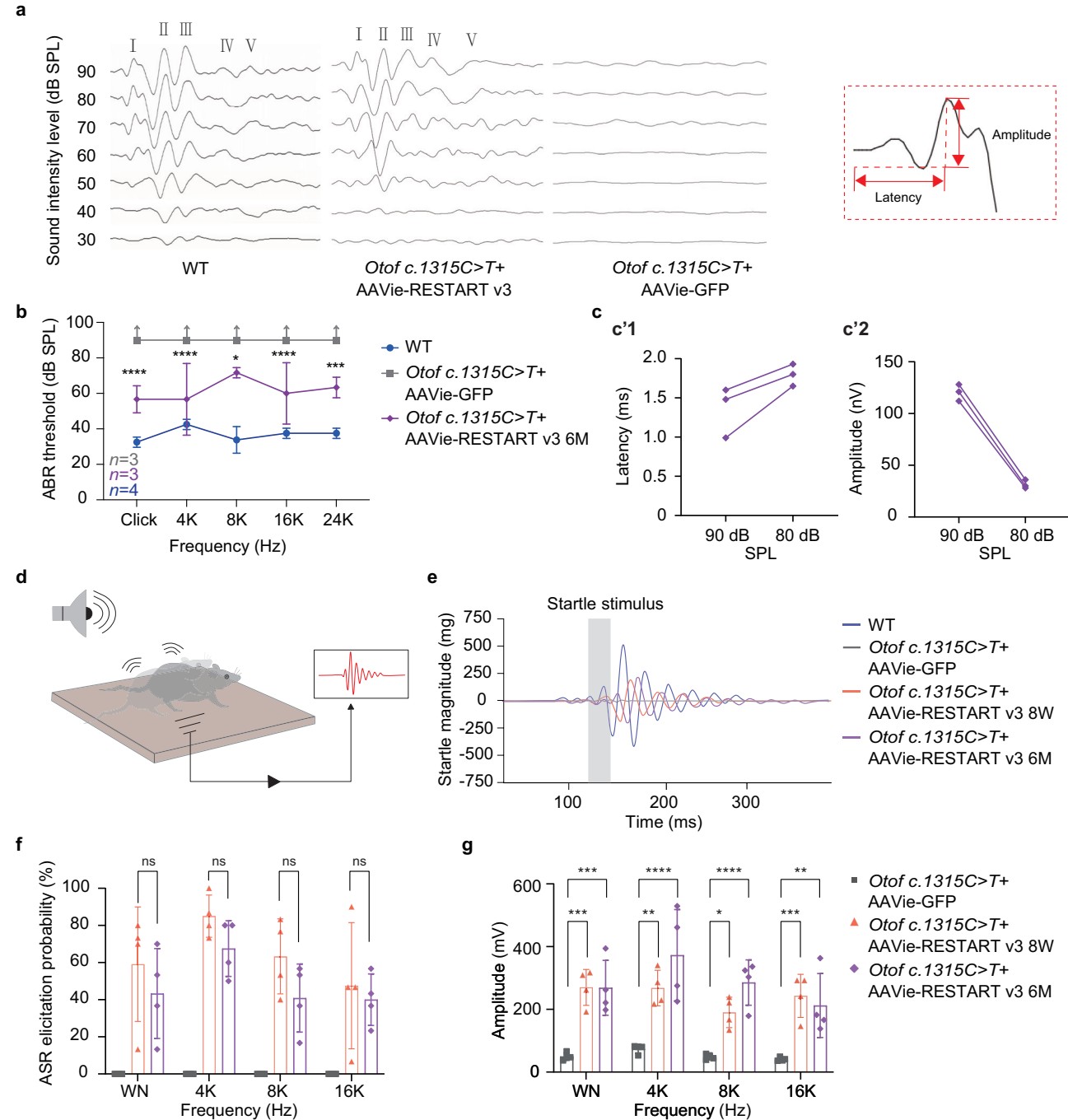

**Fig. 5 | RNA editing RESTART v3 can restore hearing in *Otof c.1315 C > T* (p.R439\*) mice. a** Representative ABR waveforms in response to sound stimuli in the WT mice and *Otof c.1315 C > T* (p.R439\*) mice treated with RNA editing AAVie-RESTART v3 and AAVie-GFP. ABR waveforms are labeled from I-V. The right panel shows ABR wave I. **b** Analysis of ABR thresholds was performed in the WT mice, the *Otof c.1315 C > T* (p.R439\*) mice treated with AAVie-RESTART v3 and the AAVie-GFP treated mice, respectively. Data are represented as the mean ± SD. Data was analyzed by two-way ANOVA. AAVie-RESTART v3 versus AAVie-GFP: click, $P = 8.576 \times 10^{-6}$; 4 K, $P = 8.576 \times 10^{-6}$; 8 K, $P = 0.0115$; 16 K, $P = 4.600 \times 10^{-5}$; 24 K, $P = 0.0002$. (WT: $n = 4$ biological samples; AAVie-RESTART v3: $n = 3$ biological samples; AAVie-GFP: $n = 3$ biological samples). **c** Analysis of the latencies (**c'1**) and amplitudes (**c'2**) of ABR wave I in the *Otof c.1315 C > T* (p.R439\*) mice treated with AAVie-RESTART v3. **d** Schematic diagram of auditory startle reflex (ASR) assay. **e** Representative ASR waveforms of the WT mice, the AAVie-RESTART v3 treated *Otof c.1315 C > T* (p.R439\*) mice (included the 8th week and 6th month after the injection), and the AAVie-GFP treated mice, respectively. **f** The comparison of ASR

elicited probability at the frequencies among the AAVie-RESTART v3 treated *Otof c.1315 C > T* (p.R439\*) mice (the 8th week and 6th month after the injection), and the AAVie-GFP treated mice. Data are represented as the mean ± SD. Data was analyzed by two-way ANOVA. AAVie-RESTART v3 injected 8th week versus 6th month: white noise (WN), $P = 0.4447$; 4 K, $P = 0.3733$; 8 K, $P = 0.2031$; 16 K, $P = 0.8295$. **g** Analysis of startle responsive amplitude in the AAVie-RESTART v3 treated *Otof c.1315 C > T* (p.R439\*) mice at 8th week and 6th month after the injection, and the AAVie-GFP treated mice, respectively. Data are represented as the mean ± SD. Data was analyzed by two-way ANOVA. AAVie-GFP versus AAVie-RESTART v3 injected 8th week: WN, $P = 0.0002$; 4 K, $P = 0.0010$; 8 K, $P = 0.0183$; 16 K, $P = 0.0007$. AAVie-GFP versus AAVie-RESTART v3 injected 6th month: WN, $P = 0.0002$; 4 K, $P = 1.7703 \times 10^{-6}$; 8 K, $P = 8.0975 \times 10^{-5}$; 16 K, $P = 0.0038$. **f, g**: AAVie-RESTART v3: 8th week: $n = 4$ biological samples; 6th month: $n = 4$ biological samples; AAVie-GFP: $n = 3$ biological samples). ns: no significant, \*$P < 0.05$, \*\*$P < 0.01$, \*\*\*$P < 0.001$, \*\*\*\*$P < 0.0001$. Source data are provided as a Source Data file.

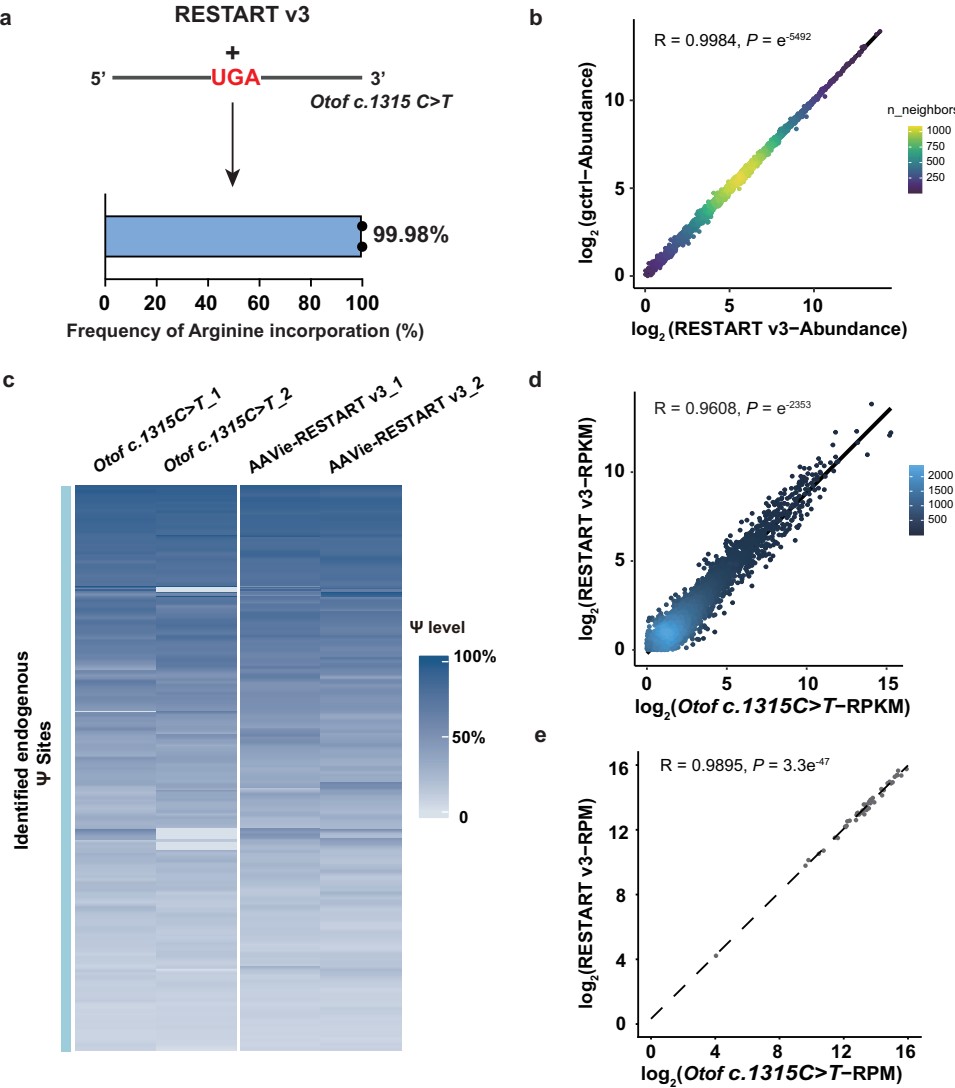

**Fig. 6 | Characterizing the specificity of RESTART v3. a** Bar plot show the arginine incorporation rate at *Otof c.1315 C > T* PTC site. *n* = 2 biological replicates. **b** Scatterplot shows the whole protein expression level by quantitative proteome analysis. Cells were transfected with the RESTART v3 or gctrl (only transfect with non-targeting control gsnoRNA). The two-sided Pearson correlation test performs correlation analysis based on the log$_2$-transformed mean Abundance values detected in two replicates of the *Otof c.1315 C > T* mice with and without RESTART v3 treatment. R represents the Pearson correlation coefficient, and the p < 0.0001 indicates a significant correlation between the two groups. **c** Heatmap showing the modification levels of identified Ψ sites across different samples. **d** Scatterplot showing the expression level of genes by RNA-seq. *Otof c.1315 C > T* (p.R439*) mice were injected with or without AAVie-RESTART v3. The two-sided Pearson correlation test performs correlation analysis based on the log$_2$-transformed mean RPKM values detected in two replicates of the *Otof c.1315 C > T* mice with and without RESTART v3 treatment. **e** Differential expression analysis of tRNA transcripts grouped by anticodons in AAVie-RESTART v3-treated cochlea relative to untreated samples. The two-sided Pearson correlation test, using log$_2$-transformed mean RPM values from two replicates, reveals a significant correlation between *Otof c.1315 C > T* mice with and without RESTART v3 treatment. Source data are provided as a Source Data file.

## The safety evaluation of RESTART v3 – AAV system

To evaluate the potential side effects of RESTART v3-AAV system in vivo, we have monitored the mice appearance characterization and body weight for long duration up to 8 months (Fig. 7a, b), we found no significant difference among WT, AAVie-GFP and AAVie-RESTART v3 treated *Otof c.1315 C > T* (p.R439*) mice groups. In this study, we have conducted four independent tests to examine possible immune reactions in response to the injection of RESTART v3-AAV system: i) The normal values of blood routine and serum biochemical examination were observed in the RESTART v3 treated groups (~6 months post-administration) (Supplementary Data 3 and 4). ii) We used RNA-seq analysis to show inflammatory factors and stress pathways in the mice cochlea, and no significant changes identified in the RESTART v3 treated group (Fig. 7c, d and Supplementary Fig. 7g, h). iii) HE staining showed no obvious side effects in the different organs in WT, AAVie-

GFP, and AAVie-RESTART v3 treated mice (~6 months post-administration, Fig. 7e; ~18 months post-administration, Supplementary Fig. 8). iv) We further detected the inflammatory level of cochlear lymph before and after the AAVie-RESTART v3 treatment using luminex liquid suspension chip, and there were no significant differences in the total 20 inflammatory cytokines in WT, AAVie-GFP and AAVie-RESTART v3 treated mice, suggesting minimal immune responses of the RESTART v3 system delivered into mice cochlea (Fig. 7f).

## Discussion

Although screening of deafness-related genes and birth defect intervention have greatly reduced the birth rate of genetically deaf children in the past decade[40–42], restoring hearing of many hereditarily deaf infants worldwide remains a challenge[3]. *Otof* is the first candidate gene to be targeted through gene therapy that resulted in complete hearing

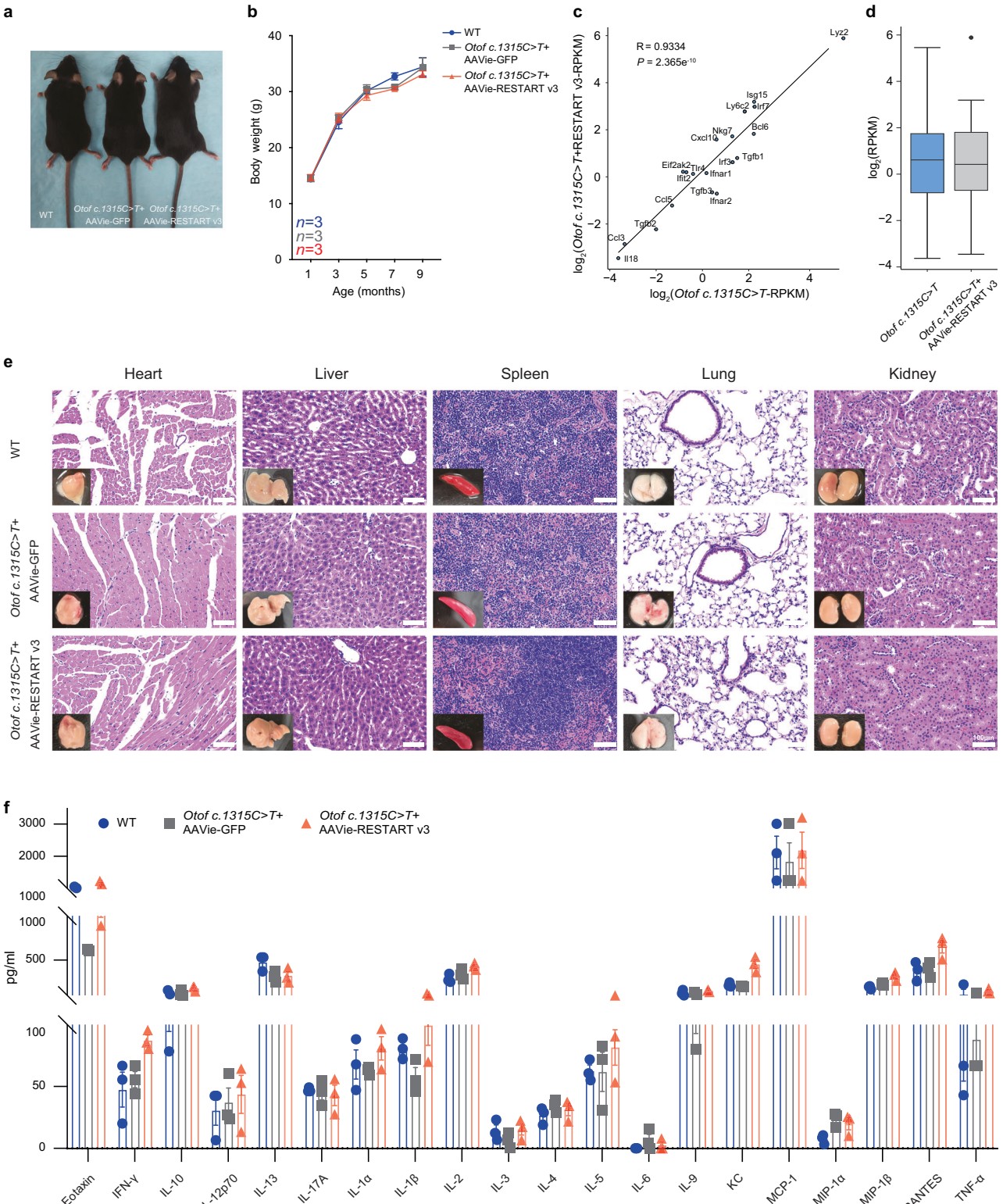

**Fig. 7 | Characterizing for RESTART v3 – AAV system safety evaluation. a** Mice appearance characterization between WT, AAVie-GFP and AAVie-RESTART v3 treated *Otof c.1315 C > T* (p.R439*) mice. **b** The body weight analysis among WT, AAVie-GFP and AAVie-RESTART v3 treated *Otof c.1315 C > T* (p.R439*) mice. *n* = 3 biological samples. Data are represented as the mean ± SD. **c** A scatter plot from transcriptome displaying the expression levels of inflammatory factors in mouse cochlear with and without RESTART v3 treatment. Statistical significance of the correlation was assessed by a two-sided Pearson test, with the strength of the association measured by the Pearson correlation coefficient. **d** Box plot showing the expression level of inflammatory factors in mouse cochlear with and without RESTART v3 treatment. The center line indicates the median, the box ends indicate the first and third quartiles. *n* = 2 biological replicates. **e** HE staining showed no side effects in different organ in WT, AAVie-GFP and AAVie-RESTART v3 treated *Otof c.1315 C > T* (p.R439*) mice (~6 months post-administration). **f** Luminex liquid suspension chip detection indicated no difference in the expression of inflammatory cytokines in WT, AAVie-GFP and AAVie-RESTART v3 treated *Otof c.1315 C > T* (p.R439*) mice. *n* = 3 biological samples. Data are represented as the mean ± SD. Source data are provided as a Source Data file.

restoration in adult congenital defect mice[18,43,44]. Moreover, the restoration of the hearing in deaf patients carrying multiple *OTOF* mutations using a double AAV vehicle has been recently reported[14], indicating that gene therapy is a valuable method for the treatment of hereditary hearing loss, and *Otof* is currently the most important target for gene therapy in the treatment of hereditary deafness[45,46].

Current gene therapies rely on the induced expression of exogenous wild-type genes in a simple and effective way[14]. However, in theory, this could lead to potential risks due to the non-physiological expression[47]. In one famous example, multiple studies have shown that replacing *SMN1* gene in treating spinal muscular atrophy (an approved gene replacement therapy indication 5 years ago) exhibited the gain of toxic function by long-term sustained overexpression of neuronal proteins SMN in the sensorimotor circuit, which is not due to the use of AAV vectors[48,49]. Hence, while the clinical trials of dual AAV-*OTOF* gene replacement therapy have achieved some initial success, they are still in the very early stage, and the long-term safety evaluation is also uncertain. In addition, it has been reported that otoferlin-expression AAV was found in organs other than the inner ear[17,21]; hence, the risk of undesired off-target expression of otoferlin, which is targeted to only be expressed in the inner ear, raises safety concerns. In contrast, RNA editing acts on endogenous mRNA transcripts, the unique in vivo expression patterns of which can improve the specificity of the RNA editing approach, and the unspecific expression beyond to cochlea was not detected in our AAVie-GFP deliver system (Supplementary Fig. 9).

Although Cas9-mediated DNA editing-based inner ear gene therapy achieves accurate manipulation in situ at specific sites in DNA[50–52], these gene editing systems also exhibits various weaknesses: i) Off-target editing. Once an off-target mutation occurs, the new mutations will be permanently introduced into the genome, leading to unpredictable biological risks[53]; ii) Reduced AAV carrying capacity; the Cas9 protein occupies most of the AAV loading capacity, leading to reduced delivering ability even when using the dual AAV strategy[15]. Compared to CRISPR-cas9 based gene editing, RNA editing is less restrained by the size of cargo (owing to the smaller size of the RNA-targeted cas systems) and does not induce permanent off-target edits.

Two recent base editing strategies have employed an adenine base editor (ABE) to treat *Otof*^{Q829X/Q829X} deafness in mice. Although the primary mutation was addressed, the therapy created a Q-to-W amino acid substitution at the target site, which might jeopardize protein function and pose a risk to future clinical applications[52,54]. In contrast, RESTART v3 minimizes the possibility of off-target effects and restores the original amino acid at the PTC site. This advantage of RESTART v3 is not limited to the *Otof c.1315 C > T* (p.R439*) mutation used in this study. According to the Human Gene Mutation Database (HGMD; www.hgmd.org), nearly half of natural nonsense mutation sites can be more likely to revert to their original amino acids using RESTART v3[29].

The clinical studies have identified >100 pathological mutations to the *OTOF* gene[34], and no hot spot *OTOF* mutation has been identified. Among the various mutations, there are more than 20 PTC-related mutations, covering over 20% of patients. Therefore, PTC-related mutations (or nonsense mutations) represent a significant group of pathological *OTOF* mutations. RESTART v3 method is a platform technology and different sgRNAs can be designed to rescue different human *OTOF* nonsense mutations. While our in vivo studies focused on a single arginine-to-stop variant in the mouse model, *Otof c.1315 C > T* (p.R439*), the added in vitro results across two additional *OTOF* PTC sites strengthen the case for broader applicability of RESTART v3 (Supplementary Fig. 10). Moreover, RESTART v3 appears to suppress NMD. Before treatment, the *Otof* transcript level in *Otof c.1315 C > T* mice was approximately 1/9 of that in WT mice; after RESTART v3 treatment, it increased to about 1/4 of WT (Supplementary Fig. 6a). In future studies, combining RESTART v3 with NMD inhibition may

further elevate *Otof* transcript levels and yield more pronounced functional improvement.

Hence, this is the in vivo demonstration of the broad applicability and potential of the RESTART system, a recently emerged RNA base editor, in treating nonsense mutation-induced hereditary hearing loss, which may be applicable for other human diseases.

## Methods

### Ethics statement
In accordance with the China National Standard GB/T 35892-2018 (Guideline for the ethical review of animal welfare), all experimental procedures were approved by the Animal Ethics Committee of Capital Medical University (AEEI-2019-169) and conducted in compliance with its standards. All possible measures were taken to alleviate suffering and to use animals sparingly.

### Plasmid construction and cell transfection
mCherry-OTOF (21 nt)-EGFP reporter construction: mCherry and EGFP coding sequences were PCR-amplified and a 21-nucleotide otof sequence was inserted between them. A 3× GGGGS linker, synthesized by Shanghai Sangon Biotechnology Company, was also included. These fragments were assembled into the pLenti-CMV-MCS backbone using NEBuilder® HiFi DNA Assembly Master Mix (New England Biolabs, E2621S) according to the manufacturer's instructions. hOTOF (R708X or R1495X)-EGFP, mOTOF (R439X)-EGFP, and gsnoRNA expression constructs: mouse *Otof* and human *OTOF* gene sequences, synthesized by Shanghai Sangon Biotechnology Company, and gsnoRNA fragments, amplified by a four-primer overlapping PCR strategy, were separately inserted into the pLenti-CMV-MCS vector and the pLenti-sgRNA backbone, respectively. U6-driven tRNA plasmid: tRNA fragments, synthesized by Beijing Xianghong Biotechnology Company, were cloned into the pLenti-sgRNA backbone. Plasmids were extracted using the EndoFree Mini Plasmid Kit II (TIANGEN BIOTECH, DP118) and transfected into HEK293T cells using Lipofectamine LTX and PLUS reagents (Invitrogen, 15338100) following the manufacturer's guidelines. The snoRNA sequences are shown in Supplementary Data 5.

### PTC readthrough analysis
48-72 hours post-transfection, cells were imaged with the ImageXpress Micro 4 high-content imaging system (Molecular Devices LLC). Sixteen images per well were captured at 10x magnification and analyzed using MetaXpress software. miRFP/mCherry-positive cells indicated successful transfection, and EGFP-positive cells indicated PTC readthrough. Readthrough efficiency was assessed according to the relative EGFP-positive cell percentage and EGFP intensity compared to a positive control.

### Quantitative off-target detection
Post-virus injection, mouse cochlear tissues were collected, and total RNA was extracted using TRIzol reagent (Invitrogen, 15596026CN). The quantitative Ψ detection was performed as the recent study described[35]. In brief, 500 ng of DNase I-treated RNA underwent rRNA removal, fragmentation, sulfite/bisulfite treatment, desalting, and desulfonation. Library construction was then performed using the SMARTer Stranded Total RNA-Seq Kit v3−Pico Input Mammalian (Takara Bio, 634485). For off-target detection under in vitro condition, RESTART v3 plasmids were transfected into HEK293T cells. After 72 hours transfection, total RNA was extracted using TRIzol reagent. Poly A + RNA was purified and performed PRAISE chemical reaction. Library construction was then performed using the SMARTer Stranded Total RNA-Seq Kit v3−Pico Input Mammalian. Library sequencing was performed on Illumina Nova 6000 with paired-end 2 × 150 bp read length.

## Targeted amplicon sequencing

Similar to the above, total RNA from cochlear tissues or transfected cells was prepared and processed. 500 ng-1000 ng full length total RNA underwent sulfite/bisulfite treatment, desalting, and desulfonation. Then the PRAISE-labeled RNA was used for reverse transcription with Maxima H minus Reverse Transcriptase (Thermo Fisher Scientific, EP0753). Next, the first round of amplicons was amplified for 20 cycles with gene-specific primers. The second round of PCR amplification was performed with the Illumina primers for 10–20 cycles. PCR products were purified using AMPure XP beads (Beckman Coulter, A63882), and size selected with polyacrylamide gel. Library sequencing was performed on Illumina Nova 6000 with paired-end 2 × 150 bp read length. The primer sequences are shown in Supplementary Data 5.

## RNA-seq and data analysis

Post-virus injection, mouse cochlear tissues were collected, and total RNA was extracted using TRIzol reagent. rRNA was removed from 100 ng DNase I treated total RNA by ribo-off rRNA depletion kit (Vazyme, N408), then was fragmented into ~150 nt. RNA was subjected to library construction using SMARTer Stranded Total RNA-Seq Kit v3–Pico Input Mammalian according to the manufacturer's protocol.

Subsequent processing was restricted to read 2. Raw sequencing data underwent quality control and adapter trimming with cutadapt (v4.2), applying key parameters: -e 0.1 -O 3 --quality-cutoff 25 -m 55. PCR duplicates were then eliminated using Seqkit[55]. The 8 bp UMI was extracted from the resulting deduplicated reads using umi_tools (v1.0.0)[56]. Furthermore, to remove the six-base constructs added to both the 5′ and 3′ ends during library preparation, umi_tools was run with the parameters: extract --bc-pattern=NNNNNNNNNNNNNNN and --3prime --bc-pattern=NNNNNN. The final cleaned reads were aligned to the mouse genome (GRCm39, NCBI Refseq) via STAR (v2.7.10b) under default settings[57]. The aligned reads were filtered with samtools (v1.14)[58] using view -f 3 -F 256. Gene-level counts were quantified by HTSeq (v1.14)[59], followed by the removal of genes exhibiting low biological reproducibility, defined as the $\log_2$ fold change (replication 1/replication 2) > 1. Finally, gene expression levels were calculated as RPKM.

## Analysis of PRAISE data

The data processing pipeline was conducted according to the established methods. In brief, subsequent analysis utilized only read 2. Initial processing of raw data involved adapter trimming and quality filtering with Cutadapt (v4.2). Subsequently, PCR duplicates were eliminated using seqkit (v0.13.2) with the command seqkit rmdup -s. A subsequent step employed umi_tools (v1.0.0) to extract and remove the 8 bp UMI from the deduplicated reads. This process, utilizing the parameters --extract-method=string --bc-pattern=NNNNNNNNNNNNNNN, also trimmed the six library-derived bases immediately following the UMI at the 5′ end of inserted sequences. Additionally, the six bases at the 3′ end were removed using umi_tools extract with the flags --extract-method=string --3prime --bc-pattern=NNNNNN.

The above processed reads were downsampled to 117 Mb and mapped via the PRAISE-tool (https://github.com/Zhe-jiang/PRAISE) to the GRCm39 reference transcriptome (NCBI Refseq). This allowed for the calculation of pseudouridine signals quantified as deletion rates. Off-target Ψ sites were subsequently identified using the following approach. (1) The deletion rate difference in *Otof c.1315 C > T* and *Otof c.1315 C > T*-AAVie RESRART v3 groups must be > 5%.(2) The deletion ratio in the PRAISE (-) samples must be <5%. (3) For off-target Ψ sites, we applied the statistical test to each of them (contingency table test between PRAISE (+) sample and PRAISE (−) sample), and the P value between the two groups must be <0.005. (4) Compared with the RESTART (−) samples, the increasing level of deletion ratio in the RESTART (+) group must be > 10%.

Based on the experimental principle, we found the potential off-target location of the whole transcriptome using bowtie (version 1.3.1)[60]. Since the target sequence was AATTTACTNAACAGAGG, we filtered potential off-target Ψ sites using the following approach: (1) The central site of the off-target sequence must be T. (2) The base after the T is random. (3) 0-3 mismatches are allowed under the first two conditions. The key parameters were: bowtie -a -v 3 -x GRCm39_trans_Refseq -c AATTTACT[ATCG]AACAGAGG.

## Analysis of Target-seq amplicon data

Following the grouping of sequencing reads by their unique molecular identifiers (UMIs), UMI groups containing fewer than 3 reads were discarded. PCR duplicates were then removed by selecting the most consensus sequence within each UMI group. The resulting deduplicated reads were subjected to adapter trimming using cutadapt (v4.2) with key parameters (-e 0.1 -O 3 --quality-cutoff 25 -m 55). UMIs were subsequently removed with umi_tools (v1.0.0). The cleaned reads were mapped to the target sequences using PRAISE-tool, and reads with more than 2 mismatches were filtered out using Samtools. Finally, the deletion rates at the target sites were calculated.

## Immunoprecipitation and peptide mass spectrometry identification

The reporter gene containing the mouse *Otof* or human *OTOF* gene sequence and flag tag was co-transfected with RESTART v3 into HEK293T cells. Two or three biological replicates were used for analysis. Seventy-two hours after transfection, the cells were lysed, and the supernatant was collected following ultracentrifugation. Flag-tagged proteins were enriched from the supernatant using anti-flag M2 conjugated magnetic beads and analyzed by coomassie staining. As previously described, the gel slices of the target size were subjected to LC-MS/MS analysis[29]. In bief, LC–MS/MS analysis was performed on EASY-nLC 1200 liquid chromatography system and Orbitrap Fusion Lumos mass spectrometer. MS data were searched in Proteome Discoverer v2.2 against a composite database comprising OTOF sequences harboring 20 alternative "X" amino acids at PTC sites, together with the UniProt Human database. Peptide-level relative abundances (PSM > 1) were quantified from precursor ion intensities in Proteome Discoverer, and the frequency of amino acid insertion at PTC site was calculated from these abundances.

## Proteomic quantitative mass spectrometry analysis

HEK293T cell samples transfected with RESTART v3 or gctrl (only transfect with non-targeting control gsnoRNA) were subjected to protein extraction using the EasyPep MS Sample Prep Kit (Thermo Fisher Scientific, A57864). Protein samples were quantified using the Pierce BCA Protein Assay Kit. Following the instructions of the EasyPep Mini MS Sample Prep Kit, 30 µg of protein was digested using Trypsin, and then mass spectrometry analysis was performed using the Orbitrap Fusion Lumos mass spectrometer. The MS raw files were processed using Proteome Discoverer 2.2 software. Search parameters specified trypsin specificity allowing up to two missed cleavages. Only peptides supported by more than one PSM were retained for analysis. Mass tolerances were set to 10 ppm for precursor ions and 0.02 Da for fragment ions. The false discovery rate was controlled at 1% at both the peptide and protein levels. Three biological replicates were used for analysis.

## tRNA-seq

Total RNA was extracted and <200 nt RNAs were enriched using the MEGAclear Transcription Clean-Up Kit (Thermo Fisher Scientific, AM1908). The isolated tRNA fraction was subjected to deacetylation by incubation with Tris 9.0 buffer at 37 °C for 45 min, followed by ethanol precipitation. Subsequently, AlkB treatment was performed to demethylate tRNA: RNA was denatured at 65 °C, incubated with a reaction

mix containing $Fe^{2+}$ iron, 2-ketoglutarate, and LAA at 37 °C for 2 h, and purified via phenol-chloroform extraction and ethanol precipitation. The RNA was then treated with T4 polynucleotide kinase (NEB, M0201S) to generate 3′-OH ends for adapter ligation. Ligation of the 3′ RNA adapter (5′-rAPP-AGATCGGAAGAGCGTCGTG-3SpC-3′) was carried out using T4 RNA ligase 2 (NEB, M0373S) at 25 °C for 2 h, followed by enzymatic digestion with 5′deadenylase (NEB, M0331S) and RecJf (NEB, M0264S) to remove excess adapter, and purification by ethanol precipitation. Reverse transcription was performed with an adapter-specific primer (ACACGACGCTCTTCCGATCT) and Induro reverse transcriptase (NEB, M0681S) at 42 °C overnight, after which excess primers were digested with Exonuclease I (NEB, M0293S). The cDNA was alkaline-denatured and purified. Ligation of the 5′ adapter (5′-Phos-NNNNNNNNNNAGATCGGAAGAGCACACGT-CTG-3SpC-3′) to the cDNA was conducted with T4 RNA ligase 1 (NEB, M0437M) at 25 °C overnight, and the product was purified again. Finally, PCR amplification was executed with index primers for library indexing, followed by purification using AMPure XP beads to generate the final tRNA-seq library. The PCR product was size-selected with 8% TBE gel. The libraries were sequenced on BGI T7 paired-end 2× 150 bp read length.

## Analysis of tRNA-seq data

Raw sequencing reads were subjected to cutadapt software (version 4.2) for quality control and adapter trimming. The key parameters were: -e 0.1 -O 3 --quality-cutoff 25 -m 15. The 10 bp UMI in the deduplication reads were removed by seqkit (version 0.13.2)[55] retaining only >20nt reads. The key parameters were: seqkit subseq -r 1:-11and seqkit seq -m 20. The mouse tRNA reference (GRCm39) was downloaded from GtRNAdb[61], followed by the addition of a CCA tail to each tRNA and the retention of a single representative sequence per isodecoder. Then the cleaned reads were mapped to the reference using bowtie2 (version 2.5.4)[62], permitting three mismatches and retaining only uniquely mapping reads. The key parameters were bowtie2 -m 3 -v 1 --best-strata. After that, we downsampled the alignments of all samples to the same level through samtools[58]. The tRNA counts were quantified by pysam (version 0.23.3). To ensure data quality, tRNAs with low biological repeatability ($\log_2$ (Replication 1/Replication 2) >1) were filtered out. Following this filtering, the expression level of each tRNA was quantified and normalized as RPM.

## Ribo-seq

Cochleae from mice injected with AAVie-RESTART v3 (3 months post-injection) and control mice were submitted to Hangzhou NeoRibo Biotechnology Co., Ltd. for Ribo-seq library preparation. Ribosome profiling was performed using a high-resolution footprinting protocol as described previously[63]. Briefly, cells were lysed under polysome-preserving conditions, and ribosome–mRNA complexes were digested with RNase I to generate ribosome-protected fragments (RPFs). RPFs were purified, dephosphorylated, ligated to adapters, and converted into cDNA sequencing libraries following the QEZ-seq protocol. Libraries were sequenced on an Illumina NovaSeq X Plus using paired-end 150 bp read length.

## Analysis of Ribo-seq data

Only read 2 was used for subsequent analysis. Raw sequencing reads were processed for adapter and quality trimming using cutadapt software (version 4.2). The key parameters were cutadapt -m 20 --quality-cutoff 20. To enrich for mRNA-derived fragments, reads aligning to rRNA and tRNA were discarded using bowtie2. The key parameters were: bowtie2 -N 0 -L 15 --end-to-end. The remaining high-quality reads were uniquely aligned to the mouse reference genome (GRCm39, NCBI Refseq or GENCODE vM38) with STAR[57], permitting 2 mismatches and retaining only uniquely mapping reads. The key parameters were STAR --outFilterType BySJout --outFilterMismatchNmax 2 --outFilterMultimapNmax 1

--quantMode TranscriptomeSAM GeneCounts. We employed the reference annotation wherein the longest coding sequence (CDS) isoform was selected for each gene. P-site positions within ribosome-protected fragments were precisely determined using the R package riboWaltz[64].

For stop codon readthrough analysis, we applied the RRTS metric, which calculates the ratio of ribosome density in the stop codon readthrough region to that in the CDS. To generate the average gene plot, ribosome density surrounding stop codons was analyzed as reported previously[65]. For individual transcripts, the density at each position was normalized against the transcript's total coding sequence read density. Translation efficiency (TE) was calculated as the ratio of gene expression measured by RNA-seq to that measured by Ribo-seq. Both expression levels were normalized using CPM. Genes included in the analysis satisfied the following criteria:

(1) they are detected in each replicate; (2) the CPM detected in RNA-seq sample is ≥ 1; and (3) the read count detected in Ribo-seq sample is ≥ 20.

## Otof c.1315 C > T (p.R439*) mouse model generation

Production of Cas9 mRNA and sgRNA. The px330 plasmid carrying the wild-type Cas9 was used as the DNA template for PCR amplification of the Cas9 coding sequence. The T7 promoter sequence was added to the forward primer and reverse primer from the coding sequence of the Cas9 gene. PCR product was amplified using the AccuPrime PCR system (Life Technologies, 12337024) and purified using the QIAquick PCR purification kit (Qiagen, 28104), in vitro transcription (IVT) was performed using the mMESSAGE mMACHINE T7 ULTRA Transcription kit (Life Technologies, AM1345). The T7-sgRNA PCR product was purified and used as the template for IVT using the MEGAshortscript T7 Transcription kit (Life Technologies, AM1354). Both the Cas9 mRNA and the sgRNAs were purified using the MEGAclear Transcription Clean-Up kit (Life Technologies, AM1908). Aliquots from an IVT reaction were separated on an agarose gel to assess the quality from a reaction. Single-stranded oligos were ordered as PAGE Ultramer from Sangon Biotech. C57BL/6 J donor female mice (3–4 weeks of age) were superovulated by administration of 5 IU (ip) of pregnant mare serum gonadotrophin (PMSG) (ProSpec HOR-272) followed 47 hrs later by 5 IU (ip) human chorionic gonadotrophin (hCG) (ProSpec HOR-272). Immediately post-administration of hCG, the female was mated 1:1 with male mouse and 22 hrs later checked for the presence of a copulation plug. Female mice displaying a copulation plug were sacrificed, the oviducts excised, and embryos collected. Standard zygote micro-injection procedure was performed on a Zeiss AxioObserver.D1 using Eppendorf NK2 micromanipulators in conjunction with Narashige IM-5A injectors. Injected zygotes were rinsed through three 30-ml drops of equilibrated KSOM (Sigma, MR-101-D) before being placed into a separate 30-ml micro drop of equilibrated KSOM where they were either cultured (Cook, MINC benchtop incubator, 37 °C, 5% $CO_2$, and 5% $O_2$/nitrogen). Two-cell embryos were processed for embryo transfer via the oviduct on the day of injection. The positive founder mice were genotyped by PCR of tail DNA and Sanger sequencing. The following primers were using for Genotyping (PrimerF: TGTTTTCCAGCTGGGGGTTT; R: TGCAAGCATTCACTTGCTTTGT).

## Gene therapy for treating Otof c.1315 C > T (p.R439*) mice

Otof c.1315 C > T (p.R439*) mice bearing a Humanized Mutation (OTOF c.1273 C > T (p.R425*)) were employed for the experimental groups, while C57BL/6 J mice from Vital River Laboratory (Beijing, China) served as controls. At four weeks postnatally, mutant mice were randomly allocated to receive either AAVie-RESTART v3 or AAVie-GFP. Randomization was not stratified by sex, as it is not a known confounding variable for the assay endpoints. All the mice were bred and housed in the Experimental Animal Department of the Capital Medical University (22–25 °C, 50% humidity, 12 hrs light/dark cycle).

## Inner ear injection

The inner ear injection was performed through the semicircular canals, as described in the previous article[66]. 4 weeks old *Otof c.1315 C > T* (p.R439*) mice were anesthetized via intraperitoneal injection of xylazine (10 mg/kg; Sigma-Aldrich, St Louis, MO, USA) and ketamine (100 mg/kg; Gutian Pharmaceutical Co., Gutian, Fujian, China). The surgeries were administered on the left ears. Following shaving and sterilization of the surgical site, the muscle overlying the temporal bone was dissected with micro-forceps to expose the posterior semi-circular canals (PSC) and lateral semicircular canals (LSC), taking care to avoid damage to adjacent vessels. A small hole was made in the mid-portion of the PSC using a 26 G needle; visible perilymphatic fluid leakage confirmed successful penetration of the bony wall. This hole was carefully enlarged to a diameter slightly larger than the polyimide tubing. The tubing tip was then gently inserted 1–2 mm into the PSC toward the crus commune. A volume of 2 μL of viral solution ($1 \times 10^{12}$ vg/mL) was injected into each ear using a microinjection pump at a rate of 0.5 μL/min. After injection, the cannula was left in place for approximately 2 minutes to allow for diffusion before being slowly withdrawn. The hole in the PSC was immediately sealed with a muscle plug, and the overlying muscle and subcutaneous tissues were repositioned. The skin incision was closed with a 4-0 suture and disinfected with povidone-iodine.

## Auditory brainstem response

The auditory brainstem response (ABR) was recorded in a pure tone audiometry room (Shengnuo Acoustic Equipment, Shanghai, China). Mice were anesthetized via intraperitoneal injection of ketamine (100 mg/kg; Gutian Pharmaceutical Co., Ltd., Fujian, China) and xylazine (10 mg/kg; Sigma-Aldrich Co., LLC, United States), with supplemental doses of ketamine (50 mg/kg) administered as needed to maintain anesthesia. The depth of anesthesia was confirmed by the absence of a toe-pinch reflex. For ABR recording, the active electrode was positioned at the midline of the anterior skull, while the reference and ground electrodes were inserted subdermally behind the test ear and the contralateral ear, respectively. A speaker was placed at the entrance of the external auditory canal of the test ear. Stimuli included tone bursts at 4, 8, 12, 16, and 24 kHz, as well as clicks (100 μs), which were delivered using System 3 hardware and SigGen/BioSig software (Tucker Davis Technologies, Alachua, FL, United States). The amplitude and latency of ABR wave I were measured at 90/80 dB sound pressure level (SPL) for each stimulus frequency. The initial stimulation intensity was set at 90 dB SPL, and the ABR threshold for each mouse was determined by gradually reducing the stimulus intensity in 10 dB steps, followed by 5 dB steps, until the lowest intensity that elicited a response was identified. The ABR threshold was defined as the minimum stimulus intensity that produced reliable and reproducible ABR waves in at least two consecutive trials.

## Auditory startle reflex

The auditory startle reflex (ASR) was measured using a startle reflex system (Beijing Macroambition S&T Co., Ltd., Beijing, China) in a pure tone audiometry room (Shengnuo Acoustic Equipment, Shanghai, China). During testing, each mouse was placed in a perforated plastic container positioned on a piezoelectric force plate. After a 5-minute silent acclimation period, acoustic stimuli were delivered through a speaker mounted above the apparatus. The stimuli included white noise and pure tones at 4, 8, and 16 kHz, all presented at 110 dB SPL. Each animal underwent a total of 120 trials—30 per stimulus type. Every trial lasted 600 ms, with the acoustic stimulus presented at 200 ms for a duration of 20 ms. The inter-trial interval was randomly varied between 10 and 20 s. The startle amplitude was quantified as the root mean square (RMS) voltage of the force plate signal immediately following stimulus onset.

## Immunofluorescence

Mice were euthanized under deep anesthesia induced by intraperitoneal injection of ketamine (100 mg/kg) and xylazine (10 mg/kg). The cochleae were then dissected under a stereomicroscope and fixed overnight at 4 °C in 4% paraformaldehyde (PFA). The following day, the samples were decalcified in 10% EDTA for 1.5 hours. After decalcification, the cochleae were permeabilized with 0.3% Triton X-100 (Sigma, USA) for 30 min and blocked in 10% goat serum (ZSGBBIO, China) for 2 h at room temperature. Subsequently, the tissues were incubated overnight at 4 °C with the following primary antibodies: rabbit anti-otoferlin (1:300, Abcam, ab309197) and mouse anti-myosin VIIa (1:300, Proteus BioSciences, 25–6790). After incubation, the cochleae were washed three times in phosphate-buffered saline (PBS), 10 min each, and then incubated for 2 h at room temperature with species-appropriate secondary antibodies conjugated to Alexa Fluor™ 488 or 594 (1:300, Invitrogen, catalog numbers A11008 and A11005). Following secondary antibody staining, the samples were washed three times in PBS, mounted on glass slides, and coverslipped using an antifade mounting medium containing DAPI (ZSGB-BIO, ZLI-9557).

## Confocal microscopy and cell counting

Confocal imaging was performed using a Leica TCS SP8 II (Wetzlar, Germany) microscope equipped with ×10/×20 air objectives and a 60× oil-immersion objective. Z-stack images were acquired in a 512 × 1024-pixel raster (x-y pixel size = 0.036 μm) with a 1-μm interval between optical planes, scanning from the apical to the basal surface of the specimen. The number of otoferlin-positive inner hair cells (IHCs) in the apical, middle, and basal turns of the organ of Corti was quantified using ImageJ software. The proportion of otoferlin-positive IHCs was calculated as the ratio of otoferlin-positive IHCs to the total number of IHCs, the latter identified by DAPI-stained nuclei.

## Routine blood test and serum chemistry in mice

Mice were anesthetized with 5% isoflurane and immediately sacrificed. Blood samples (1.0–2.0 mL) were taken from mice, which were used for routine blood test and serum chemistry. Routine blood test and serum chemistry were performed by Beijing KANG JIA HONG YUAN Biological technology co.LTD.

## Hematoxylin-Eosin staining

Tissue samples were fixed in 4% paraformaldehyde (PFA), decalcified in 0.5 M EDTA at room temperature, and dehydrated through a graded ethanol series (30%, 50%, 70%, 80%, 90%, 95%, and 100%). Following dehydration, the samples were cleared in xylene, embedded in paraffin, and sectioned at a thickness of 8–10 μm. Sections were then stained with hematoxylin and eosin (H&E). Images were acquired using a Leica DMI fluorescence microscope.

## Luminex liquid suspension chip

The assay was conducted according to manufacturer's protocol using Bio-Plex Pro Mouse Cytokine Grp (#M60009RDPD) with Luminex 200 system (Austin, TX, USA) in Wayen Biotechnologies Shanghai, Inc.

## Statistics and reproducibility

No statistical methods were used to predetermine sample sizes. Sample sizes for each experiment are indicated in the figures or figure legends and were based on established standards in the field to ensure reproducibility. Specifically, Target-seq analyses of reporter and endogenous transcripts were conducted in independent parallel experiments, with a minimum of four biological replicates. Transcriptome-wide RNA-seq and tRNA-seq analyses included two independent replicates, and Ribo-seq analyses included three independent replicates. For mouse auditory phenotyping, a minimum of three biological replicates were used. Data were included without

exclusion. While experimental samples were randomly allocated to groups, blinding was not implemented due to the involvement of multiple experimenters.

In this study, the statistical analysis was performed by GraphPad Prism 9 software (GraphPad Software Inc., La Jolla, CA, United States) for experimental data. Data were analyzed by paired/unpaired t-test for two groups, one-/two-way ANOVA for more than two groups, followed by Tukey's post hoc test or Dunn's multiple comparisons test. Data were presented as the means ± standard deviation (SD). Differences in means were considered statistically significant at $p < 0.05$, $*p < 0.05$, $**p < 0.01$, $***p < 0.001$, and $****p < 0.0001$. For the statistical analysis of sequencing data, R packages dplyr and ggpubr were used. Correlations between variables were evaluated by employing the Pearson correlation test. Other test methods follow identical principles to those employed for experimental data tests.

### Reporting summary

Further information on research design is available in the Nature Portfolio Reporting Summary linked to this article.

## Data availability

The sequence data generated in this study have been deposited in the NCBI Gene Expression Omnibus and Genome Sequence Archive, under accession code GSE262204, GSE262205, GSE262206, PRJCA049346, and PRJCA049089. The mass spectrometry proteomics data have been deposited to the ProteomeXchange Consortium via the iProX partner repository with the dataset identifier PXD070139. Source data are provided with this paper.

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

## Acknowledgements

We thank National Center for Protein Sciences at Peking University in Beijing, China, for assistance with the 4150 TapeStation System and mass spectrometry; D. Liu and Q. Zhang for their help with sample pretreatment and data analysis. We also thank the Center for Quantitative Biology at Peking University for assistance with the ImageXpress Micro 4 high-content imaging system and X. Li for her help; We thank High Performance Computing Platform of the Center for Life Science for assistance with the analysis. We thank all members of the Auditory laboratory for their many helpful discussions and support throughout this project. This work was supported by the Natural Science Foundation of China (no. 22425071 to C.Y., no. 81770997 to K.L., no. 82460223 to X.S., and no. 22407007 to H.S.), the Technology Plan Project of Beijing Tongzhou District (no. WS2024056 to K.L.), the Natural Key R&D Program of China (no. 2023YFC3402200 to C.Y.), and Beijing Municipal Science & Technology Commission (Z231100002723005 to C.Y.).

## Author contributions

K.L., C.Y., and X.S. proposed the conception of the project. H.S., W.L., Q.H., N.L., J.S., and C.Y. designed the RNA editing tools. Q.T., R.G., M.L., W.X., Q.Y., Y.L., S.G., X.S., and K.L. designed and conducted animal testing. H.S., Q.T., W.L., R.G. and M.L. collected the data and performed data analysis. H.S., Q.T., W.L., R.G. and M.L. developed photographs and schematic diagrams in the manuscript. H.S., Q.T., M.L., X.S., C.Y., and K.L. wrote the manuscript with contributions from all authors.

## Competing interests

A patent application has been filed by Peking University for the RESTART technology disclosed in this publication; C.Y. is the inventor on the patent application. The *Otof c.1315 C > T* (p.R439*) mice used in this article have obtained a Chinese patent (ZL202110894148.4); K.L. and X.S. are the inventors on this patent application. The other authors declare that they have no competing interests.
