## [Transparent Peer Review file · Nature Communications]

CRISPR-free RNA Base Editing Mediated PTC-readthrough Restores Hearing in Mice with Otof Nonsense Mutation

Corresponding Author: Professor Ke Liu

Version 1:

Reviewer comments:

Reviewer #1

(Remarks to the Author)

The revised manuscript is greatly improved and most of my concerns have been addressed properly. I have the following suggestions:

1. More in vitro tests of the RESTARTv3 in HEK293T cells are needed.
2. The authors need to examine the Otof mRNA level before and after the treatment.
3. In Fig. 3d and 3e (and some supplemental figures), still only two sets of data were presented.
4. The language needs to be further improved.

Reviewer #2

(Remarks to the Author)

This is revised study of Sun and co-authors, which I have reviewed previously. Unfortunately, I could not find evidence that the authors have incorporated the suggested experiments to assess the molecular safety of the proposed gene therapy. In their responses, the authors acknowledge the importance of these experiments and indicate a willingness to consider them. However, the revised manuscript does not include these experiments, nor does it clarify whether they are planned for future inclusion (the used wording is 'the safety evaluation is on schedule...and We believe that these experiments will more fully demonstrate the safety of the RESTART v3 system in vivo'). As I elaborate below, without this essential evidence, the study does not provide sufficiently convincing support for the in vivo suitability of the proposed gene therapy for nonsense mutations at Arg codon(s) in the OTOF gene associated with hearing loss.

In their responses, the authors felt in unnecessary polemics and comparisons between the gene replacement strategy and tRNA-Arg supplementation at gene-edited pseudoUGA PTCs. However, my review did not question the general suitability of the RESTART v3 system as a potential gene therapy approach, as I am well aware of the advantages and limitations of different strategies. Given that the authors have already demonstrated the efficacy of RESTART system in a cell model in their previous publication, this study is expected to serve as a logical continuation, focusing on establishing its in vivo suitability. To achieve this, it is essential to provide the necessary experimental evidence. I therefore reiterate the importance of the previously requested analyses:

1. Off-target effects at natural stop codons: The method of choice here would be ribosome profiling (Ribo-seq) which could precisely detect readthrough events at any of the natural stop codons or at internal PTCs utilized for selenosysteine incorporation. The sensitivity of the Ribo-seq is much higher than that of the mass spec; also it necessitates much lower amounts. The authors have performed RNA-seq, which requires similar tissue amounts as Ribo-seq.
2. Stability and expression level of tRNA-Arg-TCT: Ideally, a tRNA-seq should be the method of choice and should be performed at two or three different time points (i.e. shortly after administration, 1 month and 3-4 months). The tRNA-seq requires also similar low amounts as Ribo-seq and RNA-seq. This approach will simultaneously establish the episomal tRNA-Arg-TCT expression level (over the natural expression), but also any perturbations of the cellular tRNAome. This latter point is particularly important because, unlike suppressor tRNAs, the approach used here involves expression of a natural sense codon-decoding tRNA. This could, through feedback mechanisms, potentially alter the expression of other members of the Arg-isoacceptor family. If tRNA-seq is not feasible, the tRNA-Arg-TCT expression level from the episomal vector and

stability of the expression over the natural tRNA-Arg-TCT expression at different times postadministration should be shown using classic approaches for single tRNA detection (e.g. northern blot, qPCR etc).

3. Are any stress pathways activated? The available RNA-seq data could provide the answer and they should be further analyzed to address whether any stress pathway (integrated-stress response, immunity-associated stress, ribotoxic stress etc.) are activated after administration.

4. What is the rationale for using the AAVie system? A clinical trial for the hereditary hearing deafness has been launched and uses AAV1 vehicle. Would it be senseful to test the same vehicle with a proven safety, instead of using an entirely new AAV serotype? What is known about the safety of the AAVie serotype? Are data on this available?

5. Is the system versatile? Would it correct PTCs at another Arg codons in the OTOF transcript? It is well established that the PTC sequence context affects suppression efficacy, thus without testing the effect at different PTCs (same identity but localized in different regions of the transcript) the narrative of the paper should be adjusted to convey that it is a proof-of-principle study using a non-humanized mouse for establishing RESTART v3 effect in vivo at a single PTC at Arg codon (R439X).

Version 2:

Reviewer comments:

Reviewer #1

(Remarks to the Author)

The authors successfully addressed most of my questions in the revised manuscript. I have only two minor suggestions remained.

(1) In Fig. S6a, the authors showed that the level of Otof transcript in treated Otof mutant mice is around 1/4 of that in WT mice, which might help to explain why this treatment only partially rescues hearing in mice. The authors need discuss this.

(2) Some writing/grammar mistakes need to be fixed. Examples are listed below.

Line 126: "Otof" should not be italic and all letters should be capital.

Line 179: "in the (Fig. 5a)".

Line 219: "quantitatively" should be "quantify"; "in" should be deleted.

Line 309: "Replacement" should be "replacement".

Line 310: "Gain" should be "gain".

Line 461: "gctrl" should be "ctrl".

Reviewer #2

(Remarks to the Author)

The authors have adequately addressed my comments.

Point-by-point Response to Reviewer

Reviewer #1:

The revised manuscript is greatly improved and most of my concerns have been addressed properly. I have the following suggestions:

Response: We appreciate the reviewer's constructive comments. Please see our responses below to your specific comments. For your convenience, we have included the newly added panels immediately below our responses. Note that all of the new panels have been incorporated into the revised manuscript. Changes are highlighted in the revised manuscript with a yellow background.

1. More *in vitro* tests of the RESTARTv3 in HEK293T cells are needed.

Response: Per your request, we have performed additional *in vitro* tests in HEK293T cells to further validate the reproducibility, robustness, and versatility of RESTART v3 system. The key updates are summarized below:

(1) We added two additional biological replicates to measure PTC readthrough efficiency and on target pseudouridine (Ψ) level, so that we have four biological replicates in total. The new data consistently show that RESTART v3 enables high-efficiency PTC readthrough with precise U-to- Ψ editing. Please see the revised manuscript: Fig. 3c–e and Fig. S3b, S4a–c, and S5b.

Figure 3 (c) Top, schematics of gsnRNA-PTC-reporter, DKC1 isoform 3 (iso3) and tRNA-R-TCT-1-1 constructs. Bottom, the indicated gACA19-PTC-reporter, DKC1 iso3, and tRNA-R-TCT-1-1 were transfected into HEK293T cells. Representative fluorescence images of cells. Scale Bars, 200 μ m. This experiment was conducted with four independent replicates. **(d-e)** The indicated gACA19-PTC-reporter, DKC1 iso3, and tRNA-R-TCT-1-1 were transfected into HEK293T cells. Bar plot showing the relative fraction of EGFP positive cells **(d)**. Bar plot showing different modification level across different RESTART versions **(e)**. n = 4 biological replicates. Data are represented as the mean \pm SD.

Supplementary Figure 3 (b) The indicated gACA19-PTC-reporter, DKC1 iso3, and tRNA-R-TCT-1-1 were transfected into HEK293T cells. Bar plot showing the relative fraction of EGFP intensity. n = 4 biological replicates. Data are represented as the mean \pm SD.

Supplementary Figure 4 | gACA36 has a comparable performance to gACA19. (a-c) The gACA36-PTC-reporter, DKC1 iso3, and tRNA-R-TCT-1-1 were transfected into HEK293T cells. Representative fluorescence images of cells (**a**). Bar plot showing the relative fraction of EGFP intensity (**b**). Bar plot showing different modification level across different RESTART versions (**c**). $n = 4$ biological replicates. Data are represented as the mean \pm SD.

Supplementary Figure 5 | RESTART v3 incorporates pseudouridine modification at PTC site of full-length *Otof* mRNA. (a) The schematics of mOTOF (Full length, R439X)-PTC-reporter, gACA19, DKC1-iso3 and tRNA-R-TCT-1-1 constructs. (**b**) The indicated mOTOF (Full length, R439X)-PTC-reporter, gACA19, DKC1-iso3 and tRNA-R-TCT-1-1 were transfected into HEK293T cells. Bar plot showing different modification level across different RESTART versions. $n = 4$ biological replicates. Data are represented as the mean \pm SD.

(2) To further examine the versatility of RESTART v3, we tested two additional OTOF nonsense variants associated with human deafness (c.2122C>T, p.R708X; or c.4483C>T, p.R1495X; see Supplementary Fig. 10). Upon RESTART v3 treatment, we observed a marked increase in EGFP-positive cells, indicating successful readthrough across distinct sequence contexts. Subsequent mass spectrometry analysis confirmed near-complete fidelity (~100%) of arginine restoration at both PTC sites. Collectively, these results show the potential of RESTART as a versatile and precise platform technology for treating nonsense mutation-associated deafness.

Supplementary Figure 10 | RESTART v3 corrects other human-deafness-associated nonsense mutations within OTOF gene. (a-c) The hOTOF R708X-reporter and RESTART v3 were transfected into HEK293T cells. Representative fluorescence images of cells **(a)**. Bar plots showing the relative fraction of EGFP positive cells and intensity **(b)**. Bar plot showing the arginine incorporation rate at human OTOF (p. R708X) PTC site **(c)**. **(e-g)** The hOTOF R1495X-reporter and RESTART v3 were transfected into HEK293T cells. Representative fluorescence images of cells **(e)**. Bar plots showing the relative fraction of EGFP positive cells

and intensity (f). Bar plot showing the arginine incorporation rate at human OTOF (p. R1495X) PTC site (g). (a-g) n = 3 biological replicates. Data are represented as the mean \pm SD.

2.The authors need to examine the *Otof* mRNA level before and after the treatment.

Response: Thank you for your suggestion. As recommended, we have quantified *Otof* mRNA expression level via qRT-PCR both before and after treatment. Relative to wild-type controls, *Otof*-R439X mice exhibited a marked reduction in *Otof* transcript abundance consistent with nonsense-mediated mRNA decay (NMD). Following RESTART v3 treatment, *Otof* mRNA levels were partially restored, showing an approximately twofold increase compared with untreated *Otof*-R439X samples. These results indicate that RESTART v3 alleviates NMD-associated transcript loss and increases *Otof* mRNA abundance.

Supplementary Figure 6 (a) qRT-PCR results showing relative *Otof* expression levels in WT and *Otof* c.1315 C>T mice with or without RESTART v3 treatment. n = 4 biological samples. The statistical analyses are unpaired Student's t tests. Data are represented as the mean \pm SD.

3.In Fig. 3d and 3e (and some supplemental figures), still only two sets of data were presented.

Response: Per your request, we have added more biological replicates in revised manuscript (now n = 4; see Fig. 3c-e and Fig.S3b, S4b,c, and S5b), which we already mentioned above in our response to your Comment #1. In addition, we also added two more replicates for quantification of in vivo editing level at the PTC site (now n = 5; see Fig.4e). The results demonstrated an approximate 30% Ψ modification level at the targeted *Otof c.1315 C>T* PTC site in mouse.

Figure 4 (e) Nearly 30% pseudouridine modification efficiency achieved at the targeted PTC site in the treated *Otof c.1315C>T* (p.R439*) mice (n = 5 biological replicates).

4.The language needs to be further improved.

Response: Thank you. We have improved the language.

Reviewer #2:

This is revised study of Sun and co-authors, which I have reviewed previously. Unfortunately, I could not find evidence that the authors have incorporated the suggested experiments to assess the molecular safety of the proposed gene therapy. In their responses, the authors acknowledge the importance of these experiments and indicate a willingness to consider them. However, the revised manuscript does not include these experiments, nor does it clarify whether they are planned for future inclusion (the used wording is 'the safety evaluation is on schedule...and We believe that these experiments will more fully demonstrate the safety of the RESTART v3 system in vivo'). As I elaborate below, without this essential evidence, the study does not provide sufficiently convincing support for the in vivo suitability of the proposed gene therapy for nonsense mutations at Arg codon(s) in the OTOF gene associated with hearing loss.

In their responses, the authors felt in unnecessary polemics and comparisons between the gene replacement strategy and tRNA-Arg supplementation at gene-edited pseudoUGA PTCs. However, my review did not question the general suitability of the RESTART v3 system as a potential gene therapy approach, as I am well aware of the advantages and limitations of different strategies. Given that the authors have already demonstrated the efficacy of RESTART system in a cell model in their previous publication, this study is expected to serve as a logical continuation, focusing on establishing its in vivo suitability. To achieve this, it is essential to provide the necessary experimental evidence. I therefore reiterate the importance of the previously requested analyses:

Response: We appreciate your thorough review and detailed suggestions to our work. In this revision, we have performed and integrated key safety assessments, including: (1) ribosome profiling (Ribo-seq) to detect off-target readthrough at NTCs; (2) tRNA-seq to quantify tRNA abundance and evaluate

isoacceptor perturbations; and (3) multi-organ histopathology to assess systemic tolerance. Please see our responses below to your specific comments. For your convenience, we have included the newly added panels immediately below our responses. All of the new panels have been incorporated into the revised manuscript.

1. Off-target effects at natural stop codons: The method of choice here would be ribosome profiling (Ribo-seq) which could precisely detect readthrough events at any of the natural stop codons or at internal PTCs utilized for selenosysteine incorporation. The sensitivity of the Ribo-seq is much higher than that of the mass spec; also it necessitates much lower amounts. The authors have performed RNA-seq, which requires similar tissue amounts as Ribo-seq.

Response: We appreciate the reviewer's suggestion. Per your request, we performed ribosome profiling (Ribo-seq) on cochleae from untreated *Otof* *c.1315C>T* mice and mice treated with RESTRAT v3 for three months:

(1) Transcriptome-wide readthrough at NTCs. We first quantified ribosome-protected fragment (RPF) density within 3' UTRs downstream of annotated stop codons across all transcripts. We observed no enrichment of downstream 3' UTR RPF signal in RESTART v3–treated samples relative to untreated controls (Supplementary Fig. 7a), indicating no detectable increase in off-target readthrough at NTCs after treatment. At the per-transcript level, we computed a transcript-level Ribosome Readthrough Score (RRTS). We detected 15 and 17 potential readthrough events in untreated and treated samples, respectively. 6 events are shared between groups, while 9 and 11 events appear to be present in untreated and treated samples, respectively (Supplementary Fig. 7b). For the 11 events, we did not detect Ψ at the corresponding stop codons, and these sites lacked complementarity to the

guide snoRNA. These features argue against RESTART-mediated editing and are consistent with endogenous and low-frequency readthrough. Collectively, these analyses show no measurable increase in off-target readthrough at natural stop codons after treatment.

Supplementary Figure 7 (a) Metagene plot showing normalized reads of ribosome-protected fragments (RPFs) relative to the distance from the normal stop codon at position 0.

Supplementary Figure 7 (b) Box plot of ribosome readthrough score (RRTS) values derived from ribosome profiling of mice cochlear tissues with or without RESTART v3 treatment. RRTS values were calculated for transcripts harboring different normal stop codons, UAA, UAG and UGA. The center line indicates the median, the box ends indicate the first and third quartiles. The value of n represents the number of read-through events.

(2) Global translational homeostasis. Integrating Ribo-seq with RNA-seq, we found no significant shift in the distribution of translational efficiency after RESTART v3 treatment (Supplementary Fig. 7c). This is concordant with our quantitative proteomics, which showed no change in global protein abundance

following RESTART v3 (Fig. 6b). Together, these data indicate preserved translational homeostasis.

Supplementary Figure 7 (c) Cumulative distribution of protein-coding gene translation efficiency in *Otof c.1315 C>T* mice with and without RESTART v3 treatment.

Figure 6 (b) Scatterplot shows the whole protein expression level by quantitative proteome analysis. Cells were transfected with the RESTART v3 or gctrl.

(3) Purity of the incorporated amino acid at the PTC site. Due to low *Otof* transcript abundance, Ribo-seq coverage at the c.1315C>T PTC was insufficient for a definitive recoding analysis. To directly identify the amino acid incorporated, we performed targeted mass spectrometry. The readthrough product incorporated arginine at the PTC with 99.98% probability, indicating high-fidelity restoration (Fig. 6a).

Figure 6 (a) Bar plot show the arginine incorporation rate at *Otof c.1315 C>T* PTC site.

In summary, Ribo-seq reveals no increase in off-target readthrough at natural stop codons after RESTART v3 treatment; translational efficiency and proteome-wide abundance remain unchanged, supporting preserved translational homeostasis; and mass spectrometry at the *Otof* PTC demonstrates high incorporation fidelity. These results support the specificity and favorable molecular safety profile of RESTART v3 in the cochlea.

2. Stability and expression level of tRNA-Arg-TCT: Ideally, a tRNA-seq should be the method of choice and should be performed at two or three different time points (i.e. shortly after administration, 1 month and 3-4 months). The tRNA-seq requires also similar low amounts as Ribo-seq and RNA-seq. This approach will simultaneously establish the episomal tRNA-Arg-TCT expression level (over the natural expression), but also any perturbations of the cellular tRNAome. This latter point is particularly important because, unlike suppressor tRNAs, the approach used here involves expression of a natural sense codon-decoding tRNA. This could, through feedback mechanisms, potentially alter the expression of other members of the Arg-isoacceptor family. If tRNA-seq is not feasible, the tRNA-Arg-TCT expression level from the episomal vector and stability of the expression over the natural tRNA-Arg-TCT

expression at different times postadministration should be shown using classic approaches for single tRNA detection (e.g. northern blot, qPCR etc).

Response: Per your request, we performed tRNA sequencing (tRNA-seq) on mouse cochleae to quantify episomal tRNA-Arg-TCT expression. We profiled cochlear tRNAs at two key stages after AAVie-RESTART v3 delivery: 2 weeks (onset of AAV expression) and 6 weeks (steady-state AAV expression in the cochlea). We generated two independent replicates per condition and time point.

We first quantified the stability and expression level of the near-cognate tRNA, tRNA-Arg-TCT (R-TCT-1-1), which is one component of the RESTART v3 system. Relative to untreated controls, R-TCT-1-1 levels were elevated at 2 weeks and further increased by 6 weeks, consistent with the maturation and stabilization of AAV-driven expression. By 6 weeks, R-TCT-1-1 abundance reached approximately 1.5-fold over untreated cochleae (Supplementary Fig. 7d). This magnitude aligns with our prior cell-based studies in HEK293T cells, where exogenously expressed R-TCT-1-1 typically reached ~2-fold over wild-type cells (PMID: 38448662).

Supplementary Figure 7 (d) Expression level of tRNA-R-TCT-1-1 in the mouse cochlea before, 2 weeks (2w) after, and 6 weeks (6w) after RESTART v3 treatment. n = 3 biological replicates.

To evaluate potential feedback on the tRNA pool—particularly within the Arg isoacceptor family—we analyzed tRNA-seq data at multiple levels. When analyzed by anticodon identity, no significant changes were detected across tRNA groups after treatment (Fig.6e). Further examination, categorizing tRNAs into isodecoder families, revealed no global disturbances in tRNA expression neither (Supplementary Fig. 7e). Within the arginine isoacceptor family, only the introduced R-TCT-1-1 exhibited ~1.5-fold elevated abundance; no significant changes were observed in other Arg-isoacceptor tRNA (Supplementary Fig. 7f). These findings indicate that RESTART v3 mediates specific expression of the near-cognate tRNA without measurable perturbation to the endogenous tRNA pool, underscoring its specificity and safety.

Figure 6 (e) Differential expression analysis of tRNA transcripts grouped by anticodons in AAVie RESTART v3-treated cochlea relative to untreated samples. Data are represented as the mean value of two biological replicates.

Supplementary Figure 7 (e) Differential expression analysis of tRNA transcripts grouped by iso-decoders in AAVie RESTART v3-treated cochlea relative to untreated samples. Data are represented as the mean value of two biological replicates.

Supplementary Figure 7 (f) Expression levels of the Arg-isoacceptor family in mouse cochlear tissue, untreated and after 6 weeks of RESTART v3 treatment. n = 2 biological replicates.

3. Are any stress pathways activated? The available RNA-seq data could provide the answer and they should be further analyzed to address whether any stress pathway (integrated-stress response, immunity-associated stress, ribotoxic stress etc.) are activated after administration.

Response: We re-analyzed our RNA-seq datasets to specifically assess activation of canonical stress pathways following RESTART v3 administration, including the integrated stress response, immunity-associated stress/inflammation, ER stress, and ribotoxic stress. The results showed no obvious induction of stress-related genes after RESTART v3 treatment, further supporting the safety profile of the RESTART system. We have added related results in the revised manuscript (Supplementary Fig. 7g and h).

Supplementary Figure 7 (g) A scatter plot from transcriptome displaying the expression levels of stress-related genes in mouse cochlear with and without RESTART v3 treatment. **(h)** Box plot displaying the expression levels of stress-related genes in mouse cochlear with and without RESTART v3 treatment. The center line indicates the median, the box ends indicate the first and third quartiles.

4. What is the rationale for using the AAVie system? A clinical trial for the hereditary hearing deafness has been launched and uses AAV1 vehicle. Would it be sensible to test the same vehicle with a proven safety, instead of using an entirely new AAV serotype? What is known about the safety of the AAVie serotype? Are data on this available?

Response: To develop a safe and efficacious vector targeting inner ear cells, Zhong Guisheng's team has identified an AAV variant in 2019[1]. The creation of this variant entailed the insertion of a cell-penetrating peptide (CPP)-mimicking sequence (DGTLAVPFK), derived from the AAV PHP.eB vector, into the VP1 capsid of AAV-DJ. This AAV variant was designed to, notably target cochlear hair cells, thus was named AAV-ie (AAV-inner ear); it has been reported to infect 100% of IHCs in neonatal mice[1]. In our study, we introduced AAV-ie into 4-week-old C57BL/6J mice via the posterior semicircular canal. After 4 weeks, we found high transduction efficiency of AAV-ie in IHCs, with the rates of >95% in the apical-middle turn and >95% in

the middle-basal turn of the cochlea (Fig.4b). Because OTOF expression is required in IHCs, this efficiency advantage was decisive for an in vivo proof-of-concept.

Figure 4 (b-b'1) The images of the surgical operation of PSC; **(b-b'2)** Representative confocal microscopy image of transfection efficiency of AAVie serotype virus in inner ear, indicating a highly specific and transfected efficiency in IHCs, no immunostained signals were seen in OHCs. IHCs: inner hair cells, OHCs: outer hair cells.

Safety of AAV-ie: Published studies introducing AAV-ie into the mouse inner ear report: no adverse effects on hair-cell counts or stereocilia architecture; no significant alterations in auditory brainstem response (ABR) thresholds or vestibular performance [1–3]. In this study, we additionally followed AAV-ie–treated mice long-term. Up to ~18 months post-administration, HE staining of major organs (heart, liver, spleen, lung, kidney) revealed no abnormalities (Supplementary Fig. 8). These observations, together with unaffected global translation/proteome metrics and RNA expression level in the cochlea from our Ribo-seq/proteomics/RNA-seq analyses (reported elsewhere in the manuscript), support a favorable preclinical safety profile for AAV-ie in our setting.

Supplementary Figure 8. HE staining showed no significant differences in organ tissue structures between WT and AAVie-RESTART v3 treated *Otof c.1315C>T* (p.R439*) mice (~18 months post-administration).

We agree that AAV1 is being used in an ongoing clinical trial for OTOF-related deafness [4]. In line with the reviewer’s suggestion, we evaluated AAV1 in adult mice using the same surgical route and matched parameters that we used for AAV-ie, at a dose of 2×10^9 vg per mouse. Under these conditions, AAV1 produced suboptimal inner hair cell transduction (~45%; Response-only Fig.1), whereas Shu et al. reported ~60–94% IHC transduction with AAV1 at a substantially higher dose of 2×10^{11} vg per mouse [5]. Because in our mouse model AAV-ie achieved IHC-specific transduction with higher efficiency (up to 95%) at a 100-fold lower dose (2×10^9 vg per mouse), we selected AAV-ie for the in vivo efficacy studies. This choice maximizes the likelihood of demonstrating on-target functional rescue while enabling efficient transduction at doses compatible with our delivery paradigm.

Response-only Figure 1. AAV1 carrying EGFP (AAV1-EGFP) was delivered to the cochleae of 4-week-old mice, and IHC transfection was analyzed 4 weeks later at the apical-middle and middle-basal turns of the basilar membrane.

1. Fangzhi, Tan, Cenfeng, Chu, Jieyu, Qi et al. AAV-ie enables safe and efficient gene transfer to inner ear cells.[J] .Nat Commun, 2019, 10: 3733.
2. Liyan, Zhang, Yuan, Fang, Fangzhi, Tan et al. AAV-Net1 facilitates the trans-differentiation of supporting cells into hair cells in the murine cochlea.[J] .Cell Mol Life Sci, 2023, 80: 86.
3. Xuechun, Yang, Jieyu, Qi, Liyan, Zhang et al. The role of Espin in the stereocilia regeneration and protection in Atoh1-overexpressed cochlear epithelium.[J] .Cell Prolif, 2023, 56: e13483.
4. Jun, Lv, Hui, Wang, Xiaoting, Cheng et al. AAV1-hOTOF gene therapy for autosomal recessive deafness 9: a single-arm trial.[J] .Lancet, 2024, 403: 2317-2325.
5. Longlong, Zhang, Hui, Wang ,Mengzhao, Xun et al. Preclinical evaluation of the efficacy and safety of AAV1-hOTOF in mice and nonhuman primates.[J] .Mol Ther Methods Clin Dev, 2023, 31: 101154.

5. Is the system versatile? Would it correct PTCs at another Arg codons in the OTOF transcript? It is well established that the PTC sequence context affects suppression efficacy, thus without testing the effect at different PTCs (same identity but localized in different regions of the transcript) the narrative of the paper should be adjusted to convey that it is a proof-of-principle study using a non-humanized mouse for establishing RESTART v3 effect in vivo at a single PTC at Arg codon (R439X).

Response: We appreciate the reviewer's point regarding context dependence and the need to establish versatility beyond a single site. To address your comment, we evaluated RESTART v3 at two additional human-deafness-associated nonsense mutations in OTOF—R708X (c.2122C>T) and R1495X (c.4483C>T). Upon treatment of the RESTART v3, we observed a significant increase in EGFP-positive cells for both R708X and R1495X reporters, indicating robust readthrough efficiency across distinct transcript regions and sequence contexts (Supplementary Fig. 10). Note that

we tested only a limited set of guide snoRNAs (2–3 per site) due to revision timelines, these initial results were consistently positive. We expect that systematic guide optimization could further enhance activity. Subsequent mass spectrometry analysis confirmed near-complete arginine incorporation at both PTCs, supporting accurate amino acid restoration rather than miscoding. These data indicate that RESTART v3 can effectively suppress and correct multiple PTCs in OTOF that differ in local sequence context and position, supporting system versatility.

While our *in vivo* demonstration focuses on a single Arg PTC (R439X) in mouse model, the added *in vitro* results across two additional OTOF sites strengthen the case for broader applicability within PTCs. The new data have been incorporated into the revised manuscript.

Supplementary Figure 10 | RESTART v3 corrects other human-deafness-associated nonsense mutations within OTOF gene. (a-c) The hOTOF R708X-reporter and RESTART v3 were transfected into HEK293T cells. Representative fluorescence images of cells (**a**). Bar plots showing the relative fraction of EGFP positive cells and intensity (**b**). Bar plot showing the arginine incorporation rate at human OTOF (p. R708X) PTC site (**c**). **(e-g)** The hOTOF

R1495X-reporter and RESTART v3 were transfected into HEK293T cells. Representative fluorescence images of cells **(e)**. Bar plots showing the relative fraction of EGFP positive cells and intensity **(f)**. Bar plot showing the arginine incorporation rate at human OTOF (p. R1495X) PTC site **(g)**. **(a-g)** n = 3 biological replicates. Data are represented as the mean \pm SD.

Point-by-point Response to Reviewer

Reviewer #1:

“The authors successfully addressed most of my questions in the revised manuscript. I have only two minor suggestions remained.

(1) In Fig. S6a, the authors showed that the level of *Otof* transcript in treated *Otof* mutant mice is around 1/4 of that in WT mice, which might help to explain why this treatment only partially rescues hearing in mice. The authors need discuss this.”

Response: We thank the reviewer for the very helpful comments. We have added related discussion in “Discussion” section: “RESTART v3 appears to suppress NMD. Before treatment, the *Otof* transcript level in *Otof c.1315C>T* mice was approximately 1/9 of that in WT mice; after RESTART v3 treatment, it increased to about 1/4 of WT (Supplementary Fig. 6a). In future studies, combining RESTART v3 with NMD inhibition may further elevate *Otof* transcript levels and yield more pronounced functional improvement.”

“(2) Some writing/grammar mistakes need to be fixed. Examples are listed below.”

Line 126: “*Otof*” should not be italic and all letters should be capital.

Line 179: “in the (Fig. 5a)”.

Line 219: “quantitatively” should be “quantify”; “in” should be deleted.

Line 309: “Replacement” should be “replacement”.

Line 310: “Gain” should be “gain”.

Line 461: “gctrl” should be “ctrl”.

Response: We thank the reviewer for the very helpful comments. We have updated these mistakes.

Reviewer #2:

“The authors have adequately addressed my comments.”